# Ancient and modern anticonvulsants act synergistically in a KCNQ potassium channel binding pocket

Rían W. Manville[1] & Geoffrey W. Abbott[1]

Epilepsy has been treated for centuries with herbal remedies, including leaves of the African shrub *Mallotus oppositifolius*, yet the underlying molecular mechanisms have remained unclear. Voltage-gated potassium channel isoforms KCNQ2–5, predominantly KCNQ2/3 heteromers, underlie the neuronal M-current, which suppresses neuronal excitability, protecting against seizures. Here, in silico docking, mutagenesis and cellular electrophysiology reveal that two components of *M. oppositifolius* leaf extract, mallotoxin (MTX) and isovaleric acid (IVA), act synergistically to open neuronal KCNQs, including KCNQ2/3 channels. Correspondingly, MTX and IVA combine to suppress pentylene tetrazole-induced tonic seizures in mice, whereas individually they are ineffective. Co-administering MTX and IVA with the modern, synthetic anticonvulsant retigabine creates a further synergy that voltage independently locks KCNQ2/3 open. Leveraging this synergy, which harnesses ancient and modern medicines to exploit differential KCNQ isoform preferences, presents an approach to developing safe yet effective anticonvulsants.

[1] Bioelectricity Laboratory, Department of Physiology and Biophysics, Irvine Hall 291, School of Medicine, University of California, Irvine, CA 92697, USA. Correspondence and requests for materials should be addressed to G.W.A. (email: abbottg@uci.edu)

pilepsy affects over 50 million people worldwide, with approximately 50% being inadequately treated with currently available anti-epileptic drugs[1]. In the developing world, an estimated 80% of epilepsy patients use herbal remedies for primary healthcare[2,3]. Extract taken from leaves of the shrub *Mallotus oppositifolius* has been used traditionally in folk medicine to treat disorders such as seizures in African countries[4,5] including Ghana, where it is known as nyanyaforowa (pimpim), and Nigeria, where it is referred to as okpo-biriba. Accordingly, *M. oppositifolius* extract has been shown to delay the onset, frequency, and duration of seizures in the acute chemoconvulsant (pentylene tetrazole) mouse model[5]. Despite the clear therapeutic effects of *M. oppositifolius*, the active anticonvulsant components of this extract have remained unclear, and their molecular targets unknown. Mallotoxin (MTX; a.k.a. rottlerin) is one suggested anti-seizure component of *M. oppositifolius*. However, a plausible molecular target for MTX that would quell seizures has not previously been identified. In addition, some other plants heavily used in folk medicine also contain MTX but are not traditionally used to treat epilepsy. For example, *Mallotus philippensis*, a perennial shrub distributed in outer Himalayan lowlands, is reported to possess antifilarial, antifertility, antibiotic, anti-inflammatory, and a range of other properties but is not reportedly used to treat seizures[6]. We therefore hypothesized that MTX either is not the active anticonvulsant in *M. oppositifolius* or does not act alone.

Members of the KCNQ (Kv7) subfamily of voltage-gated potassium (Kv) channels are essential for control of cellular excitability and repolarization in a wide range of cell types. Kv channels, including the KCNQs, are composed of tetramers of α subunits each containing six transmembrane segments (S1–S6), split into a voltage-sensing domain (S1–S4) and a pore module (S5 and S6) (Fig. 1a, b)[7]. KCNQ2–5 channels—predominantly KCNQ2/3 heteromers—generate the M-current, a muscarinic-inhibited Kv current that regulates neuronal excitability[8,9]. Mutations in KCNQ2 and KCNQ3 subunits underlie various forms of epilepsy, including early infantile epileptic encephalopathy, benign familial neonatal seizures, and other miscellaneous early onset encephalopathies[10–12]. Accordingly, retigabine (RTG) (also known as ezogabine) is a first-in-class anticonvulsant that works by activating KCNQ2/3 channels, negative-shifting their voltage dependence to increase their open probability at subthreshold potentials and prevent aberrant neuronal excitability[13–18].

RTG was approved by the FDA in 2011 and was in clinical use as an add-on therapy for the treatment of partial seizures in adults with epilepsy until 2017, when it was withdrawn from the market because of side effects including blue skin discoloration and retinal pigment changes. More recently, the skin discoloration has been found to be reversible after drug discontinuation[19,20]. RTG activates all neuronally expressed KCNQ isoforms (KCNQ2–5), with a preference for KCNQ3 (ref. [21]), and in addition to epilepsy showed promise in treating disorders including anxiety, neuropathic pain, neurodegenerative disorders, cancer, inflammation, and ophthalmic diseases[22–25]. New drugs are therefore needed that share mechanistic commonalties with RTG but lack the side effects, and ideally possessing improved efficacy and/or potency, to reduce the required dosage.

Here, we report that two components of *M. oppositifolius* leaf extract, MTX and IVA, act synergistically in similar binding pockets to activate KCNQ2/3 channels and reduce tonic seizure incidence and related mortality in mice. We also demonstrate that when co-administered, MTX, IVA, and RTG voltage-independently lock KCNQ2/3 open at all voltages. We explain the molecular mechanisms underlying these synergies, which suggest a pathway for developing safer, more effective anticonvulsants.

## Results

### Multiple *M. oppositifolius* leaf compounds activate KCNQ2/3.

Igwe et al.[26] recently identified by mass spectrometry nine primary components of an ethanolic extract of the *M. oppositifolius* leaf, in addition to MTX, which was previously identified in *M. oppositifolius* bark and leaves[27] (Fig. 1c). Using two-electrode voltage-clamp electrophysiology, we screened all ten compounds for their ability to activate heterologously expressed KCNQ2/3 channels in *Xenopus laevis* oocytes, with the exception of valeric acid, which we had previously found to be inactive in this respect[28] (Fig. 1d). Compounds were screened at 100 μM, except MTX and palmitic acid (30 μM). Four components—glutaconic acid, isovaleric acid (IVA), MTX, and palmitic acid, negative-shifted the voltage dependence of KCNQ2/3 activation, as quantified using KCNQ2/3 tail currents at −30 mV immediately following channel activation at voltages between −80 mV and +40 mV (Fig. 1e, f). Three of these compounds (MTX excepted) possessed strong negative electrostatic surface potential close to a carbonyl oxygen, a property previously shown important for KCNQ2/3 activation by RTG and related synthetic anticonvulsants[29]. One component (2-mercaptophenol) positive-shifted KCNQ2/3 voltage dependence, an effect more likely to be pro- rather than anticonvulsant; the other components had little-to-no effect on KCNQ2/3 activation (Fig. 1c–f; Supplementary Figures 1–6; Supplementary Tables 1–9).

### MTX potently activates KCNQ2/3 channels.

MTX, a polyphenol, was the most potent KCNQ2/3 activator in our initial screen (Fig. 1d–f). This was in contrast to a prior report in which MTX was previously found to activate KCNQ1 (a cardiac and epithelial Kv channel) and KCNQ4, while KCNQ2, KCNQ5, and heteromeric KCNQ2/3 channels were concluded to be insensitive[30]. However, the screening in that study was performed using a cell membrane potential of +40 mV, a voltage that often fails to uncover effects of openers that operate by negative-shifting the voltage dependence of activation. Indeed, we likewise found little effect of MTX on KCNQ2/3 at +40 mV, but we observed a prominent activating effect at −60 mV and consequent −17 mV negative shift in the midpoint voltage dependence ($V_{0.5}$) of KCNQ2/3 activation (Fig. 1e, f; Fig. 2a; Supplementary Figure 3; Supplementary Table 5). At −60 mV, 100 μM MTX increased KCNQ2/3 current seven-fold; the activation $EC_{50}$ was 11.5 ± 0.2 μM (standard error of the mean, SEM) (Fig. 2b). MTX activation of KCNQ2/3 began immediately upon wash-in, plateaued at 5 min, washed out slowly, but was rapidly inhibited by KCNQ-specific blocker XE991 (Fig. 2c). MTX speeded KCNQ2/3 activation and slowed deactivation, consistent with open state stabilization and closed state destabilization (Fig. 2d; Supplementary Tables 10 and 11). MTX exerted potent effects on KCNQ2/3-dependent membrane hyperpolarization (Fig. 2e), illustrative of how MTX can KCNQ2/3-dependently dampen cellular excitability. The observed effects were KCNQ2/3-dependent: MTX had no effect on water-injected control oocytes (Fig. 2f) nor on a different-subfamily Kv channel, KCNA1 (Fig. 2g).

### MTX preferentially activates KCNQ2 channels.

With respect to homomeric M-channels, KCNQ2 exhibited the highest MTX sensitivity, with an $EC_{50}$ of 6.4 μM at −60 mV. KCNQ3* (an expression-optimized KCNQ3-A315T mutant that ensures robust currents)[31] was twofold less sensitive than KCNQ2 ($EC_{50}$, 13.0 μM), while KCNQ4 and KCNQ5 had MTX $EC_{50}$ values of 20.2 and 67.1 μM, respectively (Fig. 2h, i; Supplementary Figures 7–10; Supplementary Tables 12–15). MTX activation of homomeric KCNQs was again most effective at −60 mV (Fig. 2j). Despite relatively lower sensitivity, MTX was an effective KCNQ5 opener

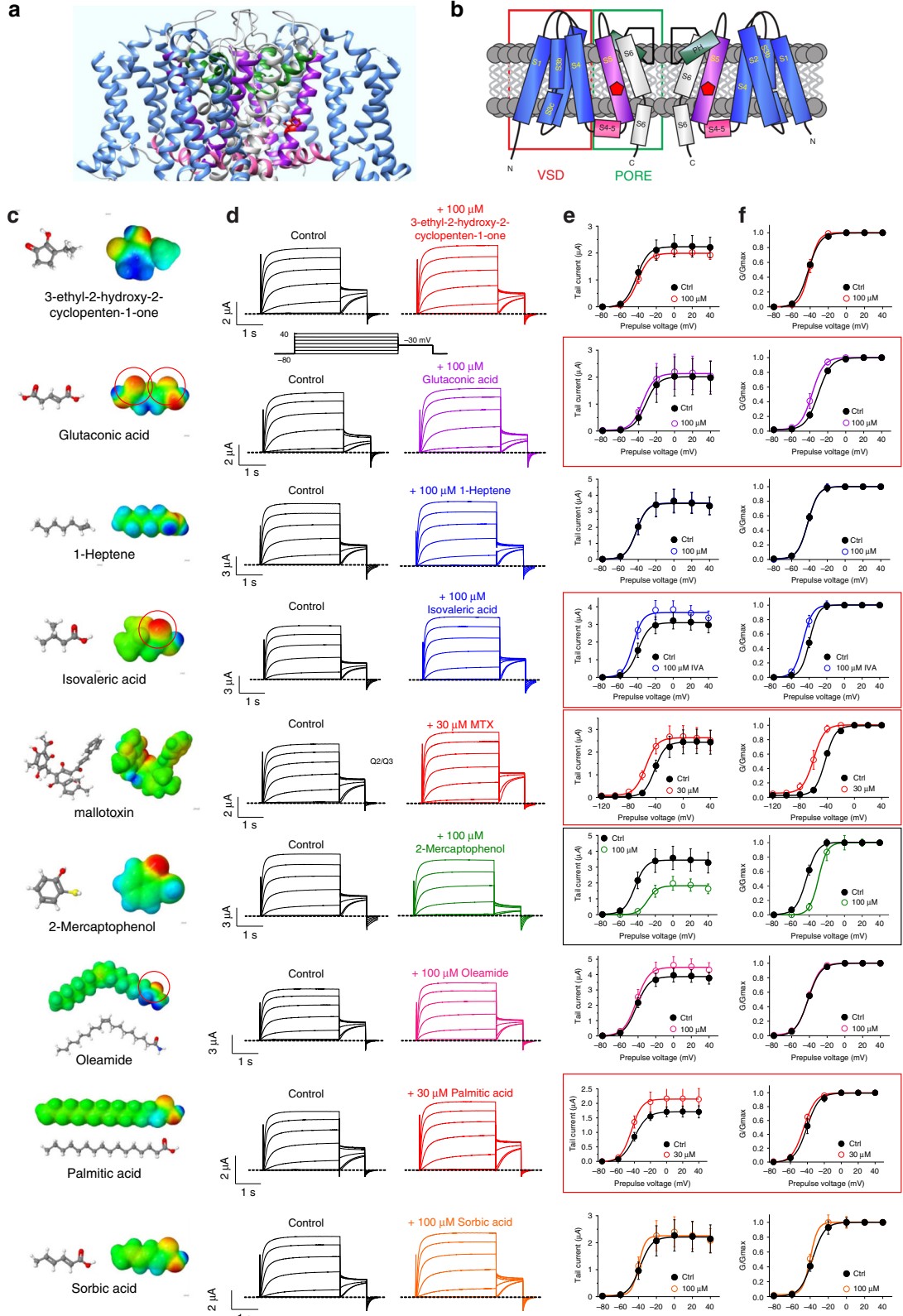

**Fig. 1** Multiple *M. oppositifolius* leaf extract compounds activate KCNQ2/3. **a** KCNQ1–KCNQ3 chimeric structure model. **b** KCNQ topology (two of four subunits shown). VSD voltage-sensing domain. **c** Structure and electrostatic surface potential (blue, positive; green neutral; red, negative) of *M. oppositifolius* leaf extract components. Open red circles highlight strongly negative electrostatic surface potential. **d** Averaged KCNQ2/3 current traces in response to voltage protocol (upper inset) when bathed in the absence (Control) or presence of *M. oppositifolius* leaf extract components ($n = 4$–16). Dashed line indicates zero current level in this and all following current traces. **e, f** Mean effects of leaf extract components (as in **d**; $n = 4$–16) on: **e** KCNQ2/3 raw tail currents at −30 mV after prepulses as indicated; **f** $G/G_{max}$. Error bars indicate SEM. Red boxes indicate KCNQ2/3 activation; black box indicates KCNQ2/3 inhibition

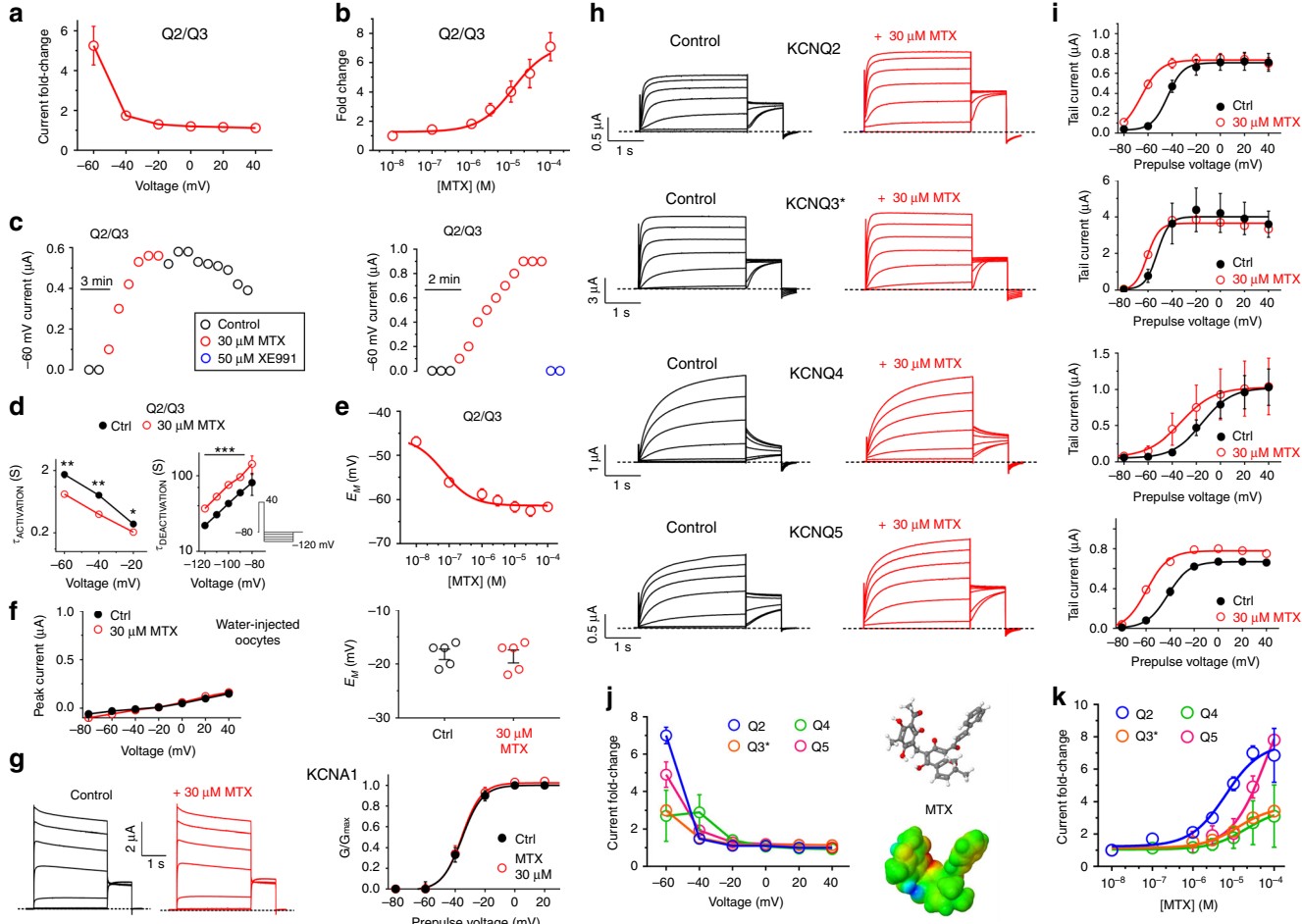

**Fig. 2** MTX preferentially activates KCNQ2. **a** Voltage dependence of KCNQ2/3 current fold-increase by MTX (30 μM), plotted from traces as in Fig. 1 ($n$ = 9). **b** Dose response of KCNQ2/3 channels at −60 mV for MTX (calculated $EC_{50}$ = 11.5 μM; $n$ = 4–9). **c** Exemplar −60 mV KCNQ2/3 current (*left*) during wash-in/washout of MTX; *right*, during wash-in of MTX followed by XE991. **d** Mean activation (*left*) and deactivation (*right*) rates for KCNQ2/3 before (Ctrl) and after wash-in of MTX ($n$ = 9); ***$p$ < 0.001. Activation rate was quantified using voltage protocol as in Fig. 1d. Deactivation rate was quantified using voltage protocol shown (*lower right inset*). **e** MTX dose-dependently hyperpolarizes resting membrane potential ($E_M$) of unclamped oocytes expressing KCNQ2/3; $n$ = 9. **f** MTX has no effect on (*left*) endogenous mean current or (*right*) $E_M$ of water-injected control oocytes ($n$ = 5). Voltage protocol as in Fig. 1d. **g** MTX has no effect on (*left*) averaged current traces or (*right*) $G/G_{max}$ of oocytes expressing KCNA1 ($n$ = 5). Voltage protocol as in Fig. 1d. **h** Averaged current traces for homomeric KCNQ2–5 channels in the absence (*Control*) or presence of MTX (30 μM) ($n$ = 5–10). Voltage protocol as in Fig. 1d. **i** Mean effects of MTX (30 μM) on −30 mV tail currents for channels and voltage protocol as in **h** ($n$ = 5–10). **j** Mean voltage dependence of 30 μM MTX (structure and surface potential, *right*) activation of homomeric KCNQ2–5 at −60 mV, recorded from tail currents as in **i** ($n$ = 5–10). **k** MTX dose response at −60 mV for homomeric KCNQ2–5, quantified from data as in **i** ($n$ = 5–10). All error bars indicate SEM

at higher concentrations, increasing −60 mV current eight-fold at 100 μM, similar to the effect on KCNQ2 at 100 μM and more than double the effect on KCNQ3* and KCNQ4 (Fig. 2k). Among the neuronal KCNQs, MTX (30 μM) had the greatest effect on the midpoint voltage of KCNQ2 activation (−24 mV), comparable to reported effects of RTG on KCNQ2 (−24 mV for 10 μM RTG[17]), followed by KCNQ2/3 and KCNQ4 (each −17 mV), KCNQ5 (−14 mV), and KCNQ3* (−9 mV) (Supplementary Tables 5 and 12–15). This order of potency is in contrast to that of RTG, which is KCNQ3 > KCNQ2/3 > KCNQ2 > KCNQ4 > KCNQ5 (refs.[17,23]).

**MTX binds close to the channel pore to activate KCNQ2/3.** RTG, an established KCNQ2/3 channel activator, requires KCNQ2-W236, which is located on transmembrane segment 5 (S5; Fig. 3a) or its equivalent on KCNQ3 (W265) for activation; these residues are thought to be required for RTG binding[32]. KCNQ2-L275 (or L314, its equivalent in KCNQ3), close to the

selectivity filter, also influences RTG activation of KCNQ2 and may impinge on or form part of the binding site[33], as illustrated here using in silico docking to a model chimeric structure derived from the *Xenopus* KCNQ1 cryo-EM structure with KCNQ3 RTG binding residues and close neighbors added (Fig. 3b, upper).

In silico docking simulations predicted binding of MTX in the region of KCNQ3-W265, but not L314 (Fig. 3b, lower), in a location we recently discovered to harbor an evolutionarily conserved neurotransmitter binding pocket in KCNQ2–5 (ref. [28]). However, previous studies clearly show that KCNQ1, which lacks the W265 equivalent, is activated by MTX[30]. This, together with our KCNQ3 docking prediction, suggested that MTX might fit in a binding pocket close enough to be influenced by W265, but not absolutely require it for binding. Electrophysiological analysis of KCNQ2 and KCNQ3 channel mutants support this hypothesis. Thus, leucine substitution of either KCNQ2-W236 or KCNQ3-W265, or both, in KCNQ2/3 channel complexes reduced as much as tenfold the potency of MTX, quantified as the negative shift in voltage dependence of activation induced, but a similar maximal

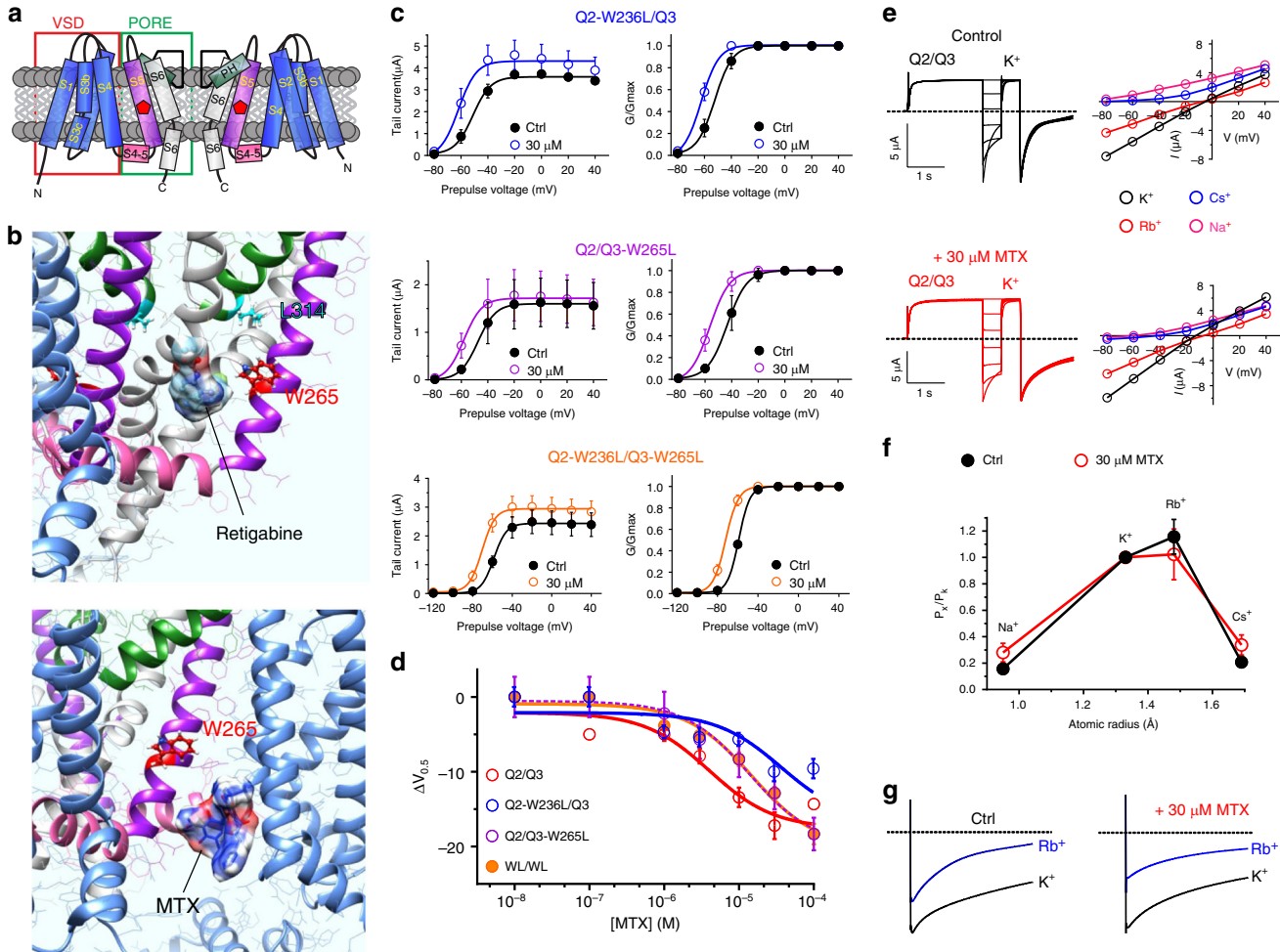

**Fig. 3** MTX activates KCNQ2/3 by binding close to the pore. **a** KCNQ topology (two of four subunits shown) indicating approximate position of KCNQ3-W265. VSD voltage-sensing domain. **b** Binding position of (*upper*) retigabine and (*lower*) MTX in KCNQ3 predicted by SwissDock using a chimeric KCNQ1–KCNQ3 structure model. **c** Effects of MTX (30 μM) on tail current and $G/G_{max}$ relationships for single- and double-W/L mutant KCNQ2/3 channels as indicated ($n = 3$–5). Voltage protocol as in Fig. 1d. **d** Dose response for mean $\Delta V_{0.5}$ of activation induced by MTX for wild-type KCNQ2/3 and mutant channels as in **c** ($n = 3$–9). **e** *Left*, exemplar traces; *right*, mean $I/V$ relationships for KCNQ2/3 channels bathed in 100 mM K$^+$, Rb$^+$, Cs$^+$, or Na$^+$ in the presence or absence (Control) of MTX (30 μM); $n = 4$–7. **f** Relative ion permeabilities of KCNQ2/3 channels in the presence or absence (Ctrl) of MTX (30 μM); $n = 4$–7. Quantified from traces and plots as in panel **e**. **g** Relative Rb$^+$ to K$^+$ permeabilities of KCNQ2/3 channels in the presence or absence (Ctrl) of MTX (30 μM); $n = 4$–8. All error bars indicate SEM

efficacy was achieved in the mutant channels (Fig. 3c, d; Supplementary Figures 11–13; Supplementary Tables 16–18).

Due to the bulky nature of MTX, we predicted that its binding within a pocket close to S5 would influence pore conformation, potentially altering relative ion permeabilities. To assess this, we first conducted pseudo-bi-ionic substitution experiments, which showed that MTX increased relative permeability of KCNQ2/3 to Na$^+$ and Cs$^+$, and decreased permeability to Rb$^+$, compared to K$^+$ (Fig. 3e, f). Secondly, we assessed the ratio of Rb$^+$ conductance to K$^+$ conductance, $G_{Rb}/G_K$, an alternative method to probe the pore conformation of K$^+$ channels (see Methods). By this method, KCNQ2/3 exhibited a baseline $G_{Rb}/G_K$ value of $0.75 \pm 0.10$; application of MTX reduced the $G_{Rb}/G_K$ value to $0.52 \pm 0.06$ (Fig. 3g). Thus, MTX binding alters KCNQ2/3 pore conformation sufficiently to alter relative ion permeabilities, and also to sense W236/W265.

In further support of this hypothesis we found that GABOB (γ-amino-β-hydroxybutyric acid), a high-affinity KCNQ2/3 partial agonist that binds close to KCNQ3-W265 (ref. [28]), only partially diminishes the effects of MTX on KCNQ2/3 channels (Fig. 4a–d; Supplementary Table 19). This is consistent with MTX binding in

a similar binding pocket to GABOB, but not directly competing for exactly the same binding site, which would result in proportionately greater inhibition of MTX effects, as we previously observed for GABOB with GABA or retigabine[28]. Interestingly, in our initial screen we found that 2-mercaptophenol inhibits KCNQ2/3 at 100 μM (Fig. 1c–e), while at lower doses it is a mild KCNQ2/3 activator (Supplementary Figure 4, Supplementary Table 6). In silico docking predicted that 2-mercaptophenol binds close to KCNQ3-W265 (Fig. 4e). Strikingly, mutation to leucine of KCNQ2-W236 and KCNQ3-W265 rendered KCNQ2/3 insensitive to 100 μM 2-mercaptophenol (Fig. 4f, g; Supplementary Table 20). At this dose, 2-mercaptophenol did not alter the ability of MTX (30 μM) to activate KCNQ2/3 channels (Fig. 4h–j; Supplementary Table 21). We conclude that despite the capacity of 2-mercaptophenol to inhibit KCNQ2/3 if administered alone, when co-administered with MTX (as in leaf extract) it cannot compete out MTX. This likely arises for two reasons: first, the 2-mercaptophenol-binding site appears to be closer to W265 (Fig. 4e–g) than that of MTX, which does not absolutely require W265 (Fig. 3). Second, in contrast to GABOB, 2-mercaptophenol

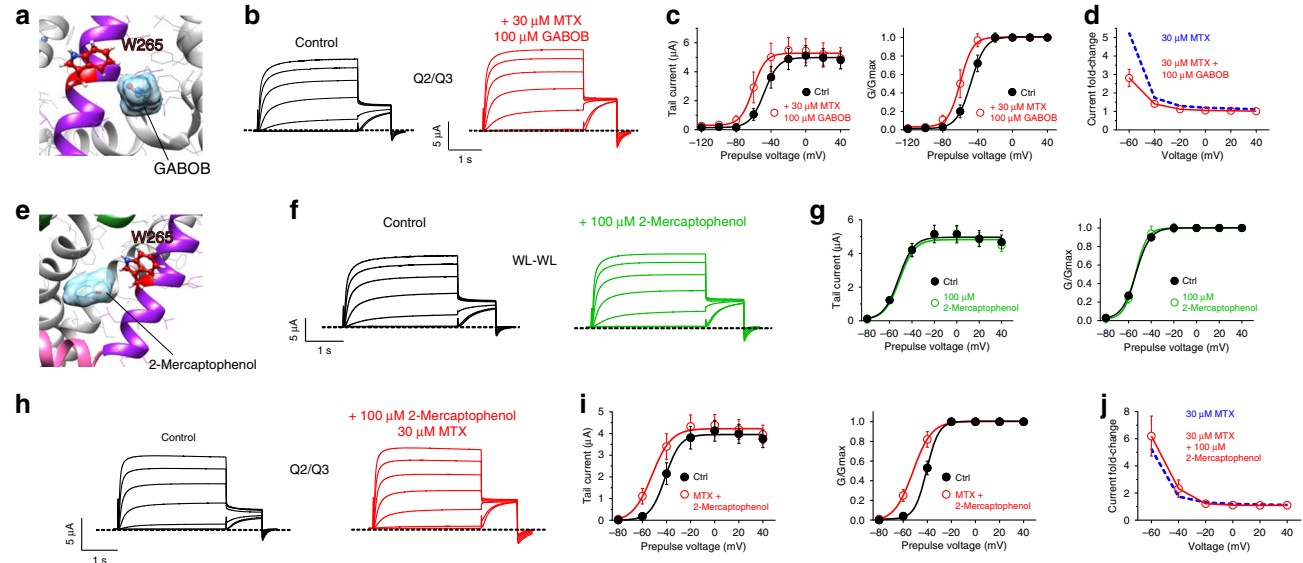

**Fig. 4** MTX outcompetes 2-mercaptophenol to activate KCNQ2/3. **a** Binding position of GABOB predicted by SwissDock using a chimeric KCNQ1–KCNQ3 structure model. **b** Exemplar traces showing effects of MTX (30 μM) with GABOB (100 μM) on KCNQ2/3 channels. Voltage protocol as in Fig. 1d. **c** Effects of MTX (30 μM) with GABOB (100 μM) on mean tail current (*left*) and $G/G_{max}$ (*right*) relationships for KCNQ2/3 ($n = 6$) calculated from traces as in panel **b**. **d** Current fold-change at −60 mV exerted by MTX (30 μM) alone (from Fig. 2a) or with 100 μM GABOB, from data as in panel **c** ($n = 6$). **e** Binding position of 2-mercaptophenol predicted by SwissDock using a chimeric KCNQ1–KCNQ3 structure model. **f** Exemplar traces showing effects of 2-mercaptophenol (100 μM) on KCNQ2-W236L/KCNQ3-W265 (WL-WL) channels. Voltage protocol as in Fig. 1d. **g** Effects of 2-mercaptophenol (100 μM) on mean tail current (left) and $G/G_{max}$ (right) relationships for KCNQ2-W236L/KCNQ3-W265 (WL-WL) channels ($n = 9$) calculated from traces as in panel **f**. **h** Exemplar traces showing effects of MTX (30 μM) with 2-mercaptophenol (100 μM) on KCNQ2/3 channels. Voltage protocol as in Fig. 1d. **i** Effects of MTX (30 μM) with 2-mercaptophenol (100 μM) on mean tail current (left) and $G/G_{max}$ (right) relationships for KCNQ2/3 ($n = 9$) calculated from traces as in panel **h**. **j** Current fold-change at −60 mV exerted by MTX (30 μM) alone (from Fig. 2a) or with 100 μM 2-mercaptophenol, from data as in panel **i** ($n = 9$). All error bars indicate SEM

is a low-affinity inhibitor, only beginning to inhibit at ~100 μM and activating at lower concentrations (Supplementary Figure 4, Supplementary Table 6).

**_M. oppositifolius_ component IVA potently activates KCNQ2.** In addition to MTX, previous gas chromatography-mass spectrometry (GCMS) analysis of _M. oppositifolius_ extract identified nine additional compounds[26], three of which we found to activate KCNQ2/3 channels in our initial screen (Fig. 1). One, IVA, is a phytocompound found both in _M. oppositifolius_ leaf extract and in the extract of valerian root (from _Valeriana officinalis_) (Fig. 5a). Valerian root has been used in herbal medicine for more than two millennia, in ancient Greece and Rome for disorders including insomnia, and since the sixteenth century in northern England and Scotland for convulsions[34,35]. As recently as 2002, an estimated 1.1% of the United States adult population (~2 million people) had used valerian root extract in the past week[36]. IVA shares a chemical feature of RTG important for KCNQ2/3 activation, i.e., strongly negative electrostatic surface potential close to a carbonyl oxygen (Fig. 5a) and is predicted by SwissDock to bind close to KCNQ3-W265 (Fig. 5b). In addition to KCNQ2/3 (Fig. 1), IVA potently and effectively activated homomeric KCNQ2 at −60 mV (EC$_{50}$, 0.34 μM) and, to a lesser extent, KCNQ3* (EC$_{50}$, 0.5 μM) and KCNQ4 (EC$_{50}$, 16.2 μM), with no effect on KCNQ5 (Fig. 5c, d; Supplementary Figures 14–17; Supplementary Tables 22–25). Like MTX, IVA did not alter currents generated by KCNA1 (Supplementary Figure 18; Supplementary Table 26).

As we also observed for MTX, IVA activation was voltage-dependent and had the greatest fold-effect on current at −60 mV (Fig. 5c, e), shifting the KCNQ2 and KCNQ2/3 activation $V_{0.5}$ by −9 and −11 mV, respectively (Supplementary Tables 4 and 22).

Unlike what we observed for MTX (Fig. 3c, d), KCNQ2-W236/ KCNQ3-W265 were essential for IVA activation of KCNQ2/3 (Fig. 5e). Thus, IVA did not increase KCNQ2/3-W236L/W265L current at −60 mV at any concentration (Fig. 5f; Supplementary Figure 19; Supplementary Tables 27 and 28). Further, the W236L/ W265L mutation prevented IVA from shifting the $V_{0.5}$ of KCNQ2/3 activation at all concentrations, in sharp contrast to the much more subtle effect of the same double mutation on KCNQ2/3 activation by MTX (Fig. 5g; Supplementary Table 29). Supporting the premise that IVA effects require the S5 tryptophan, KCNQ1 (which lacks the W) was IVA-insensitive (Fig. 5h, i), whereas KCNQ1 is activated by MTX[30]. Also in support of a binding site for IVA close to W236/W265, the partial agonist GABOB[28] was highly effective at competing out effects of IVA on KCNQ2/3 (Fig. 5j–l; Supplementary Table 30).

We also performed dose responses for the remaining KCNQ2/ 3-active compounds in _M. oppositifolius_ leaf extract, i.e., glutaconic acid, 2-mercaptophenol, and palmitic acid, and in addition tested oleamide because at 100 μM it slightly increased KCNQ2/3 currents at higher voltages (Fig. 1e). However, none of these compounds achieved the maximal efficacy we observed for IVA or MTX; as mentioned earlier, 2-mercaptophenol was inhibitory at higher doses (Fig. 5m; Supplementary Figures 1–8; Supplementary Tables 2 and 4–8).

**MTX and IVA synergistically activate KCNQ2/3 channels.** Because our data identified MTX and IVA as the most active _M. oppositifolius_ components with respect to KCNQ2/3 activation, and also suggested different binding positions for MTX and IVA (Figs. 3b and 5b), we next tested their effects in combination. MTX (30 μM) and IVA (500 μM) in combination strongly activated KCNQ2/3 current, especially at −60 mV, and shifted the

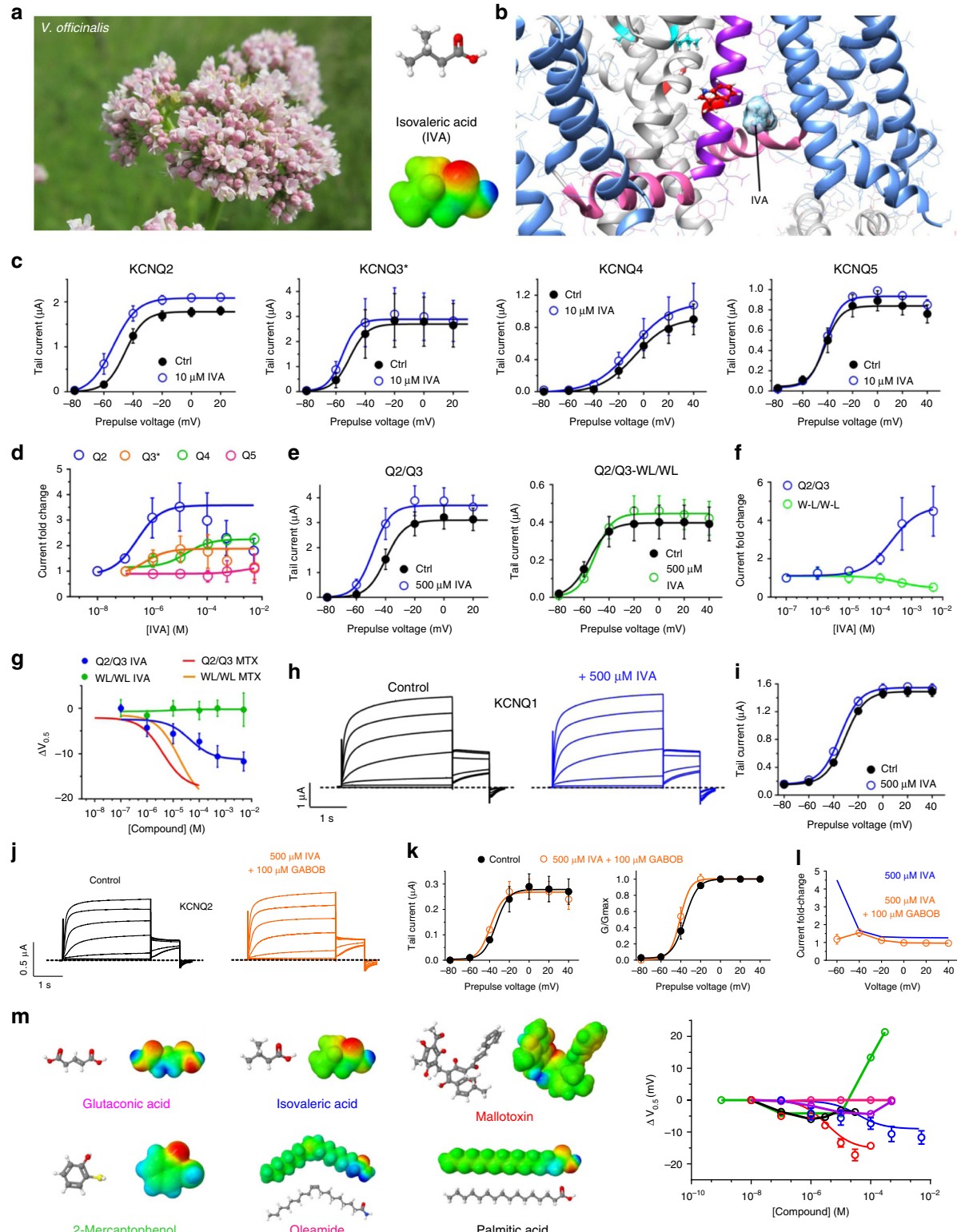

$V_{0.5}$ of activation by $-23$ mV (Fig. 6a–c; Supplementary Table 31). MTX and IVA in combination speeded KCNQ2/3 activation and slowed its deactivation >twofold (Fig. 6d; Supplementary Tables 32 and 33). Strikingly, the IVA + MTX combination synergistically increased KCNQ2/3 current at $-60$ mV by 24-fold, compared to ~fivefold for either component alone (Fig. 6e). A cocktail of the five KCNQ2/3-active components of *M. oppositifolius* leaf extract produced a similar KCNQ2/3 $V_{0.5}$ activation shift to that of IVA + MTX, again suggesting these two

as the most active and synergistic components (Fig. 6f–i). Further, in silico docking studies suggested that IVA and MTX could fit together in the W265-proximal binding pocket, providing a possible mechanistic basis for their synergy (Fig. 6j).

**MTX and IVA synergistically protect against seizures.** The cellular electrophysiology data therefore predicted that MTX and IVA would in combination be necessary and sufficient to confer

**Fig. 5** IVA activates neuronal KCNQs with preference for KCNQ2. **a** *Left, Valeriana officinalis. Right*, structure (*upper*) and electrostatic surface potential (*red, negative; blue, positive*) (*lower*) of isovaleric acid (IVA). **b** Binding position of IVA in KCNQ3 predicted by SwissDock using a chimeric KCNQ1–KCNQ3 structure model. **c** Mean tail current versus prepulse voltage relationships recorded by TEVC in *Xenopus laevis* oocytes expressing homomeric KCNQ1–5 channels in the absence (black) and presence (blue) of IVA ($n = 4$–7). Voltage protocol as in Fig. 1d. **d** IVA dose response at −60 mV for KCNQ2–5, quantified from data as in **c** ($n = 4$–7). **e** Mean tail current versus prepulse voltage relationships for wild-type KCNQ2/3 (*left*) or KCNQ2-W236L/KCNQ3-W265L (*right*) channels in the absence or presence of IVA as indicated ($n = 4$–6). Voltage protocol as in Fig. 1d. **f** Dose response for current increase at −60 mV in response to IVA for channels as in **e**. **g** Dose response for the $V_{0.5}$ of activation shift induced by IVA versus MTX in wild-type KCNQ2/3 versus KCNQ2-W236L/KCNQ3-W265L (WL/WL) channels. IVA data ($n = 4$–6) quantified from **e**; MTX data from Fig. 3d. **h** Averaged traces for KCNQ1 in the absence or presence of IVA (500 μM); $n = 6$. **i** Mean data from traces as in **h**. **j** Exemplar traces showing effects of IVA (500 μM) with GABOB (100 μM) on KCNQ2/3 channels. Voltage protocol as in Fig. 1d. **k** Effects of IVA (500 μM) with GABOB (100 μM) on mean tail current (left) and $G/G_{max}$ (right) relationships for KCNQ2/3 ($n = 5$) calculated from traces as in panel **j**. **l** Current fold-change at −60 mV exerted by IVA (500 μM) alone (from panel **f**) or with 100 μM GABOB, from data as in panel **k** ($n = 5$). **m** *Right*, dose responses for the shift in $V_{0.5}$ of KCNQ2/3 activation induced by the leaf extract compounds shown on left, calculated from traces as shown in Figs 1, 2 and 5 ($n = 4$–16). All error bars indicate SEM

anticonvulsant activity, if KCNQ2/3 activation was the molecular basis for this therapeutic action of the leaf extract. In mouse pentylene tetrazole (PTZ) chemoconvulsant assays, MTX (20 mg/kg) halved the clonic seizure incidence whereas IVA (20 mg/kg) had no effect. At 10 mg/kg neither compound reduced clonic seizures alone, but halved seizure incidence in combination (Fig. 6k). MTX and IVA were only effective at reducing tonic seizure incidence when applied in combination (halving incidence at 10 + 10 mg/kg; Fig. 6l). Most strikingly, MTX and IVA only increased survival in the seizure assay when administered in combination (tripling survival, compared to vehicle, at 10 + 10 mg/kg; Fig. 6m). Thus, MTX and IVA act synergistically to reduce seizures and seizure-related mortality in mice, mirroring their effects on KCNQ2/3 activation.

**MTX and IVA combine with RTG to lock KCNQ2/3 open.** Our results indicate that MTX binds close to the channel pore and senses the KCNQ2/3 S5 tryptophans, while IVA absolutely requires them for binding, and that KCNQ2 is the most MTX- and IVA-sensitive isoform (Figs. 2–6). Previous studies showed that in contrast, KCNQ3 is the most RTG-sensitive isoform[17,23]. These findings suggested that RTG might synergize with MTX and/or IVA. Accordingly, while RTG (10 μM) negative-shifted the KCNQ2/3 activation $V_{0.5}$ by −13 mV, RTG (10 μM) + IVA (500 μM) increased the $\Delta V_{0.5}$ to −32 mV, and RTG (10 μM) + MTX (30 μM) produced a $\Delta V_{0.5}$ of −57 mV. Most strikingly, the combination of all three compounds at these concentrations locked KCNQ2/3 open, such that its activation was voltage-independent from −120 mV to +40 mV, an effect to our knowledge not previously reported for KCNQ2/3 with any other drugs (Fig. 7a–d; Supplementary Table 34).

Kv channel openers are generally more effective at negative membrane potentials because the lower open probability provides more capacity for augmentation before the maximum open probability is reached (in contrast to positive voltages). However, at extremely hyperpolarized membrane potentials, the capacity of channel openers such as RTG to activate diminishes again (creating a bell-shaped voltage dependence to activation). In the case of KCNQ2/3 this may be because of an inability to open the channel from more stable closed conformations. However, addition of MTX or IVA, and in particular both, to RTG overcame this, resulting in potent current augmentation even at −120 mV. Thus, MTX + IVA + RTG increased KCNQ2/3 current by 60–80-fold at −80 to −120 mV (Fig. 7e).

MTX and IVA have been tolerated as part of herbal medicine for centuries[27,34,35]. RTG was in clinical use for 6 years before being withdrawn because of adverse off-target effects, although these are now known to subside following RTG discontinuation[19,20]. Given the synergy between MTX, IVA, and RTG, we tested whether combining the three at low

concentrations could achieve efficacy at potentially tolerable RTG doses. At 1 μM, RTG had negligible effects on KCNQ2/3 (Fig. 7f), as we also observed for 1 μM MTX (Fig. 2b) and 1 μM IVA (Fig. 5f). In contrast, the combination of 1 μM concentrations of RTG, MTX, and IVA was a highly effective KCNQ2/3 opener, shifting the $V_{0.5}$ of activation by −18 mV; a further negative shift in voltage dependence was observed with 1 μM RTG, 10 μM MTX, and 10 μM IVA (Fig. 7f; Supplementary Table 35). The synergy was especially apparent when comparing the fold-change in current at −60 mV induced by low doses of RTG, MTX, and IVA applied alone versus in combination (Fig. 7g, h).

We next further tested whether the heteromeric composition of KCNQ2/3 channels afforded greater sensitivity to the synergistic effects of IVA + MTX + RTG than for the homomers. Studies of relative ion permeabilities for heteromeric versus homomeric channels revealed greater increases in relative Na$^+$ and Cs$^+$ permeability (Fig. 8a–c, upper) compared to effects of MTX alone (Fig. 3f), suggesting that the triple-drug combination exerted greater effects than MTX alone on KCNQ2/3 pore conformation. However, the triple-drug combination also induced similar Na$^+$ and Cs$^+$ permeability relative to K$^+$ in homomeric KCNQ2 and KCNQ3 channels (Fig. 8a–c, middle and lower). Thus, heteromerization likely did not confer the ability to adopt a unique pore conformation not accessible to homomeric KCNQ2 or KCNQ3.

We therefore next tested whether homomeric KCNQ channels were as comprehensively activated by MTX + IVA + RTG as was KCNQ2/3. KCNQ3* was the most sensitive of the homomers, and the slowest deactivating at −120 mV of all the homomers in response to MTX + IVA + RTG. Interestingly, even homomeric KCNQ4 and KCNQ5 were activated by the highest triple-drug dose (Fig. 9a–d; Supplementary Tables 36–39). Finally, we compared the capacity of MTX + IVA + RTG to hold open the most sensitive homomer (KCNQ3*) versus heteromeric KCNQ2/3, at −120 mV (Fig. 9e). While KCNQ2/3 deactivation was minimal across 25 s, KCNQ3* current decayed >80% within 10 s (Fig. 9f).

Thus, the MTX + IVA + RTG combination leverages the heteromeric composition of KCNQ2/3 channels to exert optimal synergistic effects on channel opening. Data from Figs. 7–9 suggest that a similar pore conformation is achieved for both homomers and heteromers in the presence of MTX + IVA + RTG, but that in KCNQ2/3 this conformation is stable at more negative voltages than it is for homomeric channels.

## Discussion

We have discovered that IVA and MTX, two components of the traditional African anticonvulsant *M. oppositifolius* leaf extract, synergistically activate KCNQ2/3 and protect against tonic seizures and associated mortality. MTX, the principal component of

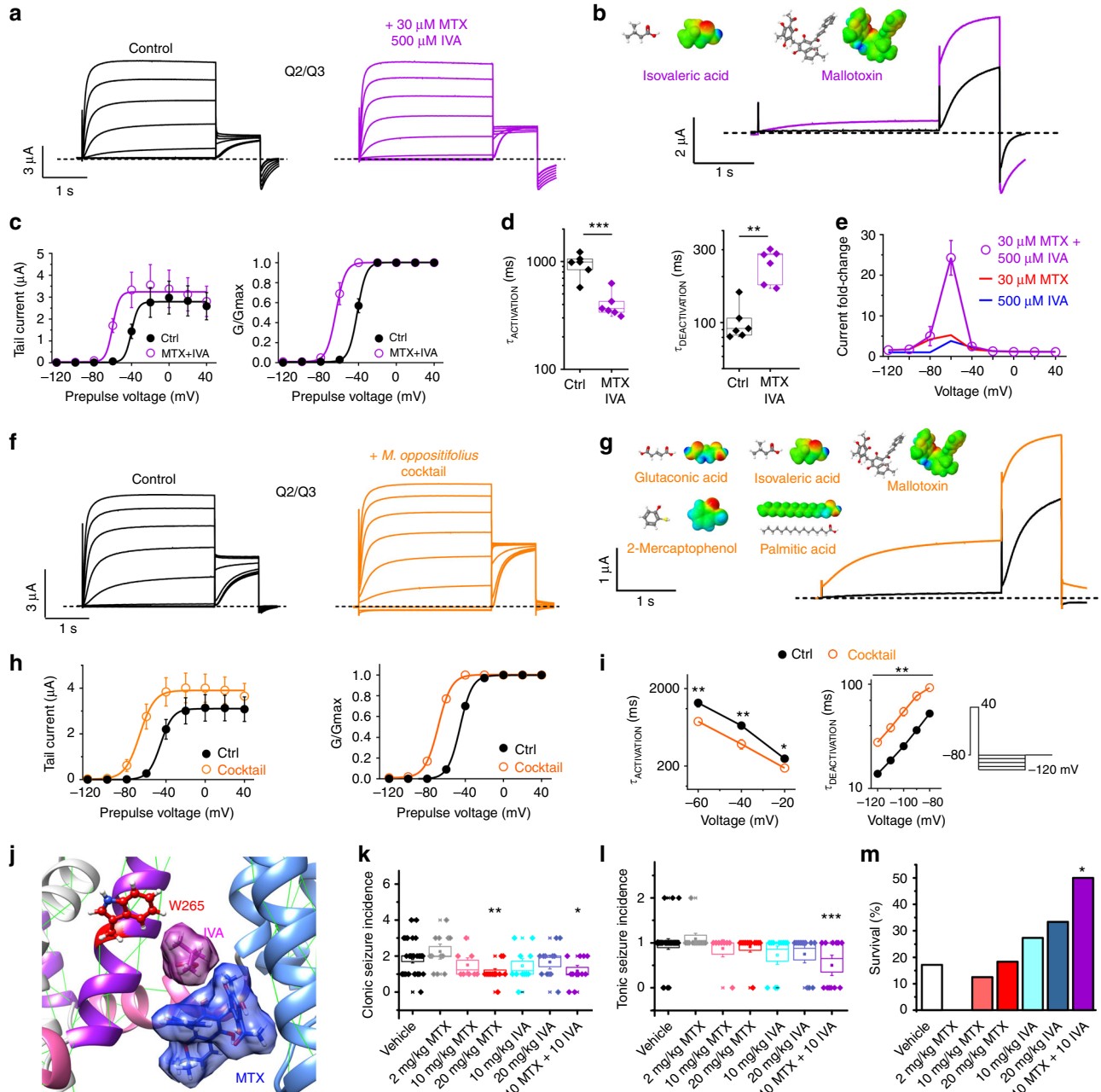

**Fig. 6** MTX and IVA synergize to activate KCNQ2/3 and protect against seizures. **a** Averaged traces showing effects of IVA and MTX on KCNQ2/3 ($n = 5$). Voltage protocol as in Fig. 1d. **b** Effects at −60 mV highlighted, from traces as in **a**. **c** Mean tail current and $G/G_{max}$ from traces as in **a** ($n = 5$). **d** Mean effect of IVA (500 μM) + MTX (30 μM) on KCNQ2/3 activation at +40 mV and deactivation at −80 mV ($n = 5$). ***$p = 0.0009$; **$p = 0.001$. **e** Mean KCNQ2/3 current fold-increase versus voltage induced by IVA and MTX alone (from Figs. 2 and 4) or in combination (from traces as in **a**); $n = 4$–9. **f** Averaged traces showing effects of leaf extract cocktail (compounds shown in **g**) on KCNQ2/3 ($n = 7$). Voltage protocol as in Fig. 1d. **g** Effects at −60 mV highlighted, from traces as in **f**. **h** Mean tail current and $G/G_{max}$ from traces as in **f** ($n = 7$). **i** Mean effect of leaf extract cocktail on rates of KCNQ2/3 activation (left) and deactivation (center; voltage protocol on right) ($n = 7$). *$p < 0.05$; **$p < 0.01$. **j** Binding position of IVA and MTX in KCNQ3 predicted by SwissDock using a chimeric KCNQ1–KCNQ3 structure model. **k**–**m** Effects of vehicle ($n = 35$) compared to IVA and MTX alone or in combination ($n = 11$–12) on **k** clonic seizure incidence, **l** tonic seizure incidence, and **m** seizure assay survival in a mouse PTZ chemoconvulsant assay. *$p < 0.05$; **$p < 0.01$; ***$p < 0.001$. Survival statistical analysis by chi-squared, all others by one-way ANOVA. All error bars indicate SEM. All box and whisker plots: box range, 25–75%, coefficient 1; whisker range, 5–95%, coefficient 1.5

phenolic extracts of *Mallotus*, has other reported biological activities;[30],[37–39] none of them readily explain anticonvulsant efficacy, but may contribute to efficacy in other therapeutic uses of *M. oppositifolius*, which include treatment of pain, infection, and inflammation[40]. Historical medicinal usage of *Mallotus* plants spans West Africa (*M. oppositifolius*) and also parts of

Asia, including Bangladesh, China, and India (*M. repandus, M. philippinensis*, and others)[6],[41]. Oral bioavailability of MTX in rats fed *Mallotus philippensis* extract was previously quantified at >20%, and plasma concentrations exceeded 2 μg/ml, reflecting also the high concentration of MTX in *Mallotus*; e.g., the MTX content of powder prepared from *Mallotus philippensis* fruit was

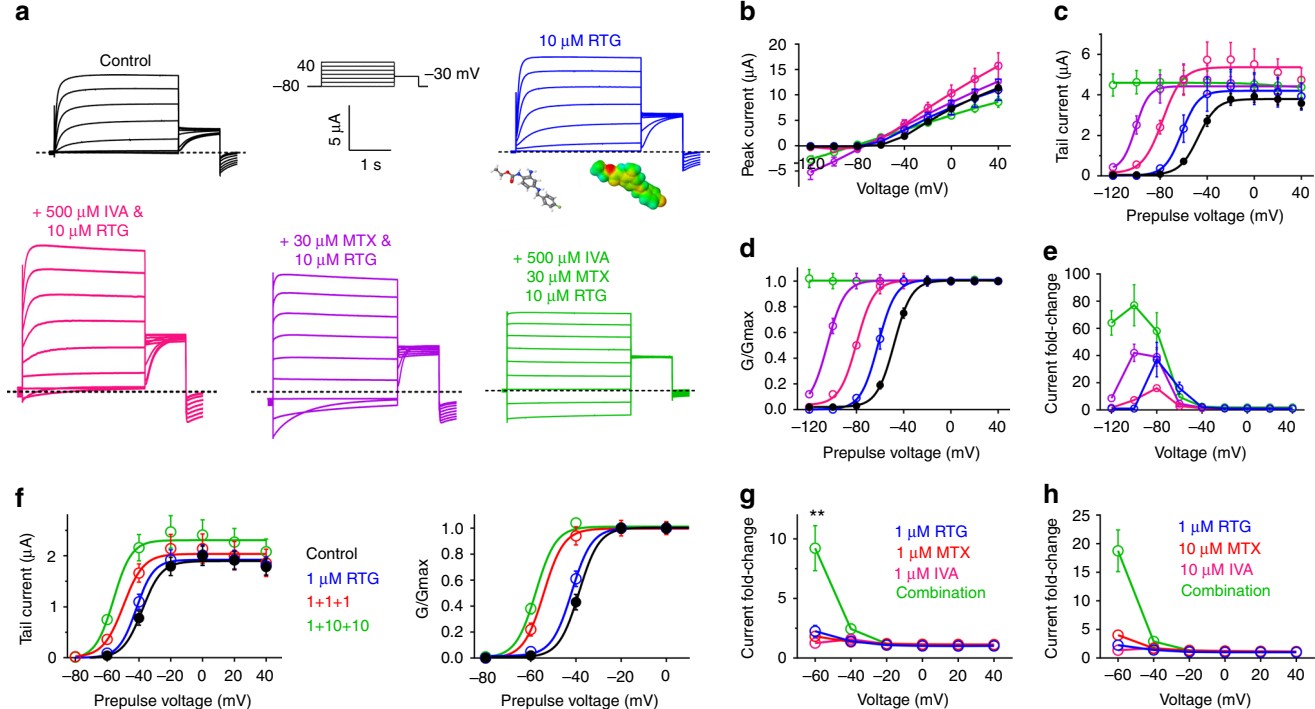

**Fig. 7** IVA and MTX synergize with RTG to lock open KCNQ2/3. **a** Averaged traces showing effects of high-dose RTG, IVA, and MTX alone or in combination on KCNQ2/3 ($n = 5$–31). Voltage protocol upper inset. **b**–**e** Analysis of traces as in **a**: **b** peak current; **c** tail current; **d** $G/G_{max}$; **e** current fold-change versus voltage; compounds and combinations color-coded as in **a**. $n = 5$–31. **f** Mean effects of low-dose 1 μM RTG versus 1 μM of each of RTG, IVA, and MTX ($1 + 1 + 1$) versus 1 μM RTG + 10 μM IVA + 10 μM MTX ($1 + 10 + 10$) on KCNQ2/3 tail currents and $G/G_{max}$ versus prepulse voltages; $n = 8$–13. **g** KCNQ2/3 current fold-increase versus voltage induced by compounds as indicated alone or in combination; $n = 8$–13. **p < 0.01. **h** KCNQ2/3 current fold-increase versus voltage induced by compounds as indicated alone or in combination; $n = 8$–13. **p < 0.01. All error bars indicate SEM. All comparisons by one-way ANOVA

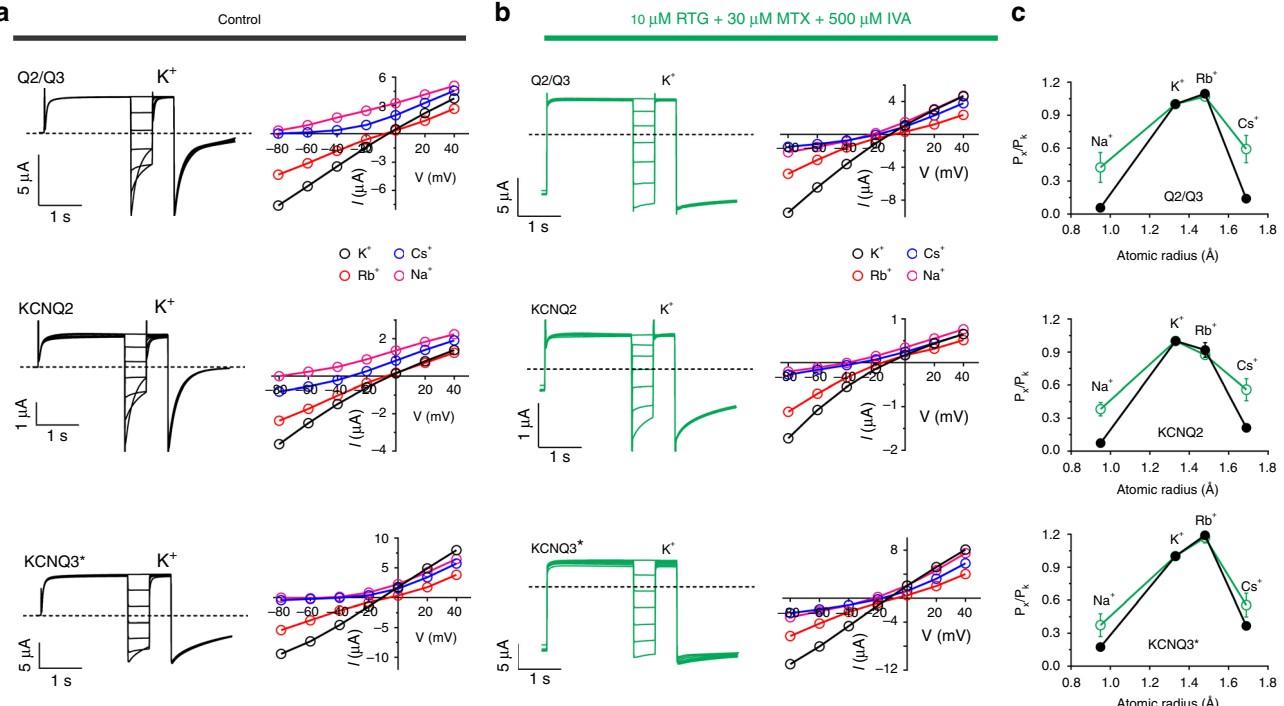

**Fig. 8** RTG + MTX + IVA alter the pore conformation of KCNQ2 and KCNQ3. **a**, **b** *Left*, exemplar traces; *right*, mean $I/V$ relationships for KCNQ2/3 (Q2/Q3) or homomeric KCNQ2 or KCNQ3 channels as indicated, bathed in 100 mM K⁺, Rb⁺, Cs⁺, or Na⁺ in the (a) absence or (b) presence of RTG (10 μM) + MTX (30 μM) + IVA (500 μM); $n = 6$–7. **c** Relative ion permeabilities of KCNQ2/3 channels in the presence (green) or absence (black) of RTG (10 μM) + MTX (30 μM) + IVA (500 μM); $n = 6$–7. Quantified from traces and plots as in panels **a**, **b**. All error bars indicate SEM

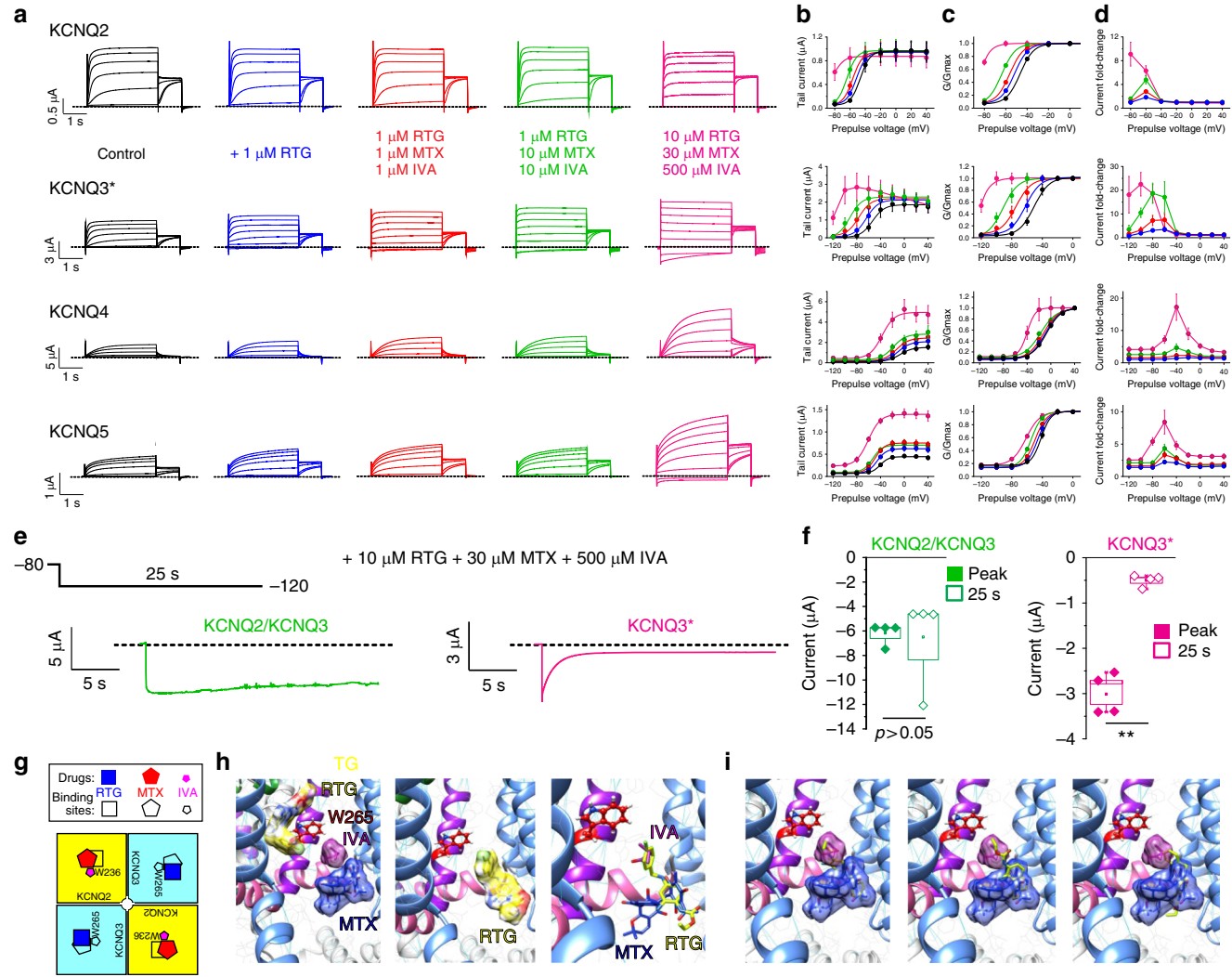

**Fig. 9** Leveraging heteromeric channel composition to lock open KCNQ2/3. **a** Averaged traces showing effects of RTG, IVA, and MTX alone or in combination, doses as indicated, on homomeric KCNQ2–5 channels ($n = 4$-11). Voltage protocol as in Fig. 1d. **b–d** Analysis of traces as in **a**: **b** tail current; **c** $G/G_{max}$; **d** current fold-change versus voltage; compounds and combinations color-coded as in **a**. $n = 4$-11. **e, f** Effects of high-dose RTG + MTX + IVA on KCNQ2/3 versus KCNQ3* held at −120 mV for 25 s. **e** Representative traces; **f** mean peak (0.5 s) versus steady-state (25 s) current. **p < 0.01; $n = 4$. Box and whisker plots: box range, SEM, coefficient of 1; whisker range 5–95%, coefficient of 1.5. **g** Model summarizing findings. Squares represent subunits within tetrameric KCNQ2/3 channels (*yellow*, KCNQ2; *pale blue*, KCNQ3). **h** Possible distinct binding positions of RTG, IVA, and MTX in one binding site in KCNQ3 (left) versus a lower-positioned RTG binding site (center) that would overlap with binding sites for IVA and RTG (right), predicted by SwissDock using a chimeric KCNQ1–KCNQ3 structure model. *Red*, KCNQ3-W265; *magenta*, IVA; *blue*, MTX; *yellow*, RTG. Space-filling omitted from molecules in *right panel* for clarity. **i** Further possible poses RTG (*yellow, no spacefill*) that would overlap with IVA (*magenta*) and MTX (*blue*) in KCNQ3 chimera model as predicted by SwissDock. All error bars indicate SEM. All comparisons by one-way ANOVA

quantified as 21.25% w/w[42]. A plasma concentration of 2 μg/ml is equivalent to 4 μM MTX, a concentration at which we observe KCNQ2/3 activation even by MTX alone.

IVA has been quantified to be ~12% of the methanol extract of the *Mallotus* leaf extract[26]. IVA is also a component of valerian root, an herbal medicine used since ancient Greek and Roman times to treat insomnia, and since medieval times in Europe specifically to treat seizures[34,35]. Valerian root extract is still used extensively today for anxiety and insomnia, although randomized controlled trials evaluating its efficacy have achieved mixed results[43–45]. It has been estimated that 10 g of valerian root might yield as much as 100 mg of IVA, and that valerian root doses of 30–50 g per day would have the potential for anticonvulsant activity[46]. Pharmacokinetic studies in humans have been performed for NPS 1776, or isovaleramide, the amide derivative of IVA. Isovaleramide readily passes though biological membranes,

and is well tolerated in humans up to at least 2400 mg per day. Absorption is rapid (mean $T_{max}$ of 30–45 min) and mean elimination half-life is 2.5 h[47]. Furthermore, volatile fatty acids similar to IVA, e.g., acetate, readily cross the blood brain barrier[48]. Taken together, these studies together with our current findings suggest that herbal extracts contain sufficient bioavailable IVA to exert effects on KCNQ2/3 channels, particularly if synergizing with MTX.

Mutagenesis results indicating differential requirement of W236/W265 (Fig. 5g) suggest that MTX and IVA can sit together in the same binding pocket (Fig. 6j), a site we recently discovered evolved to accommodate neurotransmitters including GABA[28]. When combined with the modern anticonvulsant RTG, the herbal extracts IVA and MTX have the unprecedented capacity to lock open KCNQ2/3 at all voltages tested, converting it into a voltage-independent leak channel. Our cellular electrophysiology

and docking predictions support a mechanism for this: while RTG is a more potent agonist of KCNQ3 (refs. [17,23]), IVA and MTX are more potent agonists of KCNQ2 (Figs. 2 and 5). Thus, the combination of either herbal component with RTG can leverage subunit heterogeneity in KCNQ2/3, a feature lacking in modern anticonvulsants, to increase synergy.

The data suggest a model in which MTX and IVA are able to bind to KCNQ2/3 across a wide range of membrane potentials and induce a stable open conformation. One likely configuration, and potentially the dominant configuration given the aforementioned subunit-specific potencies, is that in KCNQ2/3 heteromers MTX and IVA bind to KCNQ2, and RTG to KCNQ3 (Fig. 9g), but all combinations are considered possible. In docking studies, RTG adopted two main sets of poses near W265—one high (Fig. 9h, left) and one low (Fig. 9h, center). The former could theoretically accommodate all three molecules in one binding site, whereas the latter could not as RTG adopts essentially the same space as IVA + RTG in this pose (Fig. 9h, right). Indeed, RTG more often adopted poses in which it overlapped with separately docked IVA and MTX (more examples shown in Fig. 9i), supporting the premise that in KCNQ2/3 channels, bound IVA + MTX might prevent RTG binding to KCNQ2, but the converse might occur in KCNQ3 (Fig. 9g).

The molecular strategy we present, utilizing a combination of small molecules from ancient and modern therapeutics, may provide a route to safer, more effective anticonvulsants. Through dual synergies it lowers the effective doses required to achieve similar KCNQ2/3 opening, increases the maximal effects, and also considerably broadens the voltage range across which the drugs are effective (Figs. 7 and 9e, f).

## Methods

**Channel subunit cRNA preparation and oocyte injection.** cRNA transcripts encoding human KCNA1, KCNQ1, KCNQ2, KCNQ3, KCNQ4, or KCNQ5 were generated by in vitro transcription using the T7 polymerase message machine kit (Thermo Fisher Scientific), after vector linearization, from cDNA sub-cloned into plasmids incorporating $X.$ $laevis$ β-globin 5′ and 3′ UTRs flanking the coding region to enhance translation and cRNA stability. cRNA was quantified by spectrophotometry. Mutant cDNAs were generated by site-directed mutagenesis using a QuikChange kit according to manufacturer's protocol (Stratagene, San Diego, CA) and corresponding cRNAs prepared as above. Defolliculated stage V and VI $X.$ $laevis$ oocytes (Ecocyte Bioscience, Austin, TX) were injected with Kv channel α subunit cRNAs (10 ng total per oocyte). Oocytes were incubated at 16 °C in Barth's saline solution (Ecocyte) containing penicillin and streptomycin, with daily washing, for 3–5 days prior to two-electrode voltage-clamp (TEVC) recording.

**Two-electrode voltage-clamp.** TEVC recording was performed at room temperature with an OC-725C amplifier (Warner Instruments, Hamden, CT) and pClamp10.2 software (Molecular Devices, Sunnyvale, CA) 3–5 days after cRNA injection as described in the section above. Oocytes were placed in a small-volume oocyte bath (Warner) and viewed with a dissection microscope. Chemicals were sourced from Sigma. Bath solution was (in mM): 96 NaCl, 4 KCl, 1 MgCl$_2$, 1 CaCl$_2$, 10 HEPES (pH 7.6). Isovaleric acid, 2-mercaptophenol, 1-Heptene, and 3-ethyl-2-hydroxy-2-cyclopenten-1-one were stored at 4 °C as 5 mM stocks in Ringer's solution. MTX (DMSO), sorbic acid (ethanol), and glutaconic acid (molecular grade H$_2$O) were stored at −20 °C as 1 M stocks. Oleamide was stored as a 1 mM stock in ethanol at 4 °C. Palmitic acid was conjugated with bovine serum albumin as a 1 mM stock and stored at −20 °C. All compounds were diluted to working concentrations each experimental day. All compounds were introduced to the recording bath via gravity perfusion at a constant flow of 1 ml/min for 3 min prior to recording. Pipettes were of 1–2 MΩ resistance when filled with 3 M KCl. Currents were recorded in response to pulses between −80 and +40 mV at 20 mV intervals, or a single pulse to +40 mV, from a holding potential of −80 mV, to yield current–voltage relationships, current magnitude, and for quantifying activation rate. TEVC data analysis was performed with Clampfit10.2 (Molecular Devices) and Graphpad Prism software (GraphPad, San Diego, CA, USA); values are stated as mean ± SEM. Normalized tail currents were plotted versus prepulse voltage and fitted with a single Boltzmann function:

$$g = \frac{(A_1 - A_2)}{\left\{1 + \exp\left[\frac{V_{\frac{1}{2}} - V}{V_s}\right]\right\} y + A_2}, \tag{1}$$

where $g$ is the normalized tail conductance, $A_1$ is the initial value at $-\infty$, $A_2$ is the final value at $+\infty$, $V_{1/2}$ is the half-maximal voltage of activation, and $V_s$ the slope factor. Activation and deactivation kinetics were fitted with single-exponential functions.

For relative permeability studies, currents were recorded in response to a single pulse to +40 mV for 5 s, followed by pulses between −80 mV and +40 mV at 20 mV intervals, from a holding potential of −80 mV, to yield a current–voltage relationship. According to the Goldman–Hodgkin–Katz (GHK) voltage equation:

$$E_{rev} = \frac{RT/F \ln\left(P_K[K^+]_O + P_{Na}[Na]_O + P_{Cl}[Cl]_i\right)}{\left(P_K[K^+]_i + P_{Na}[Na]_i + P_{Cl}[Cl]_o\right)}, \tag{2}$$

where $E_{rev}$ is the absolute reversal potential and $P$ is the permeability. This permits calculation of the relative permeability of each ion if concentrations on either side of the membrane are known. A modified version of this equation was used here to determine relative permeability of two ions in a system in which only the extracellular ion concentration was known. Thus, relative permeability of Rb$^+$, Cs$^+$, and Na$^+$ compared to K$^+$ ions was calculated for all channels by plotting the $I/V$ relationships for each channel with each extracellular ion (100 mM) and comparing them to that with 100 mM extracellular K$^+$ ion to yield a change in reversal potential ($\Delta E_{rev}$) for each ion compared to that of K$^+$. Permeability ratios for each ion compared to K$^+$ were then calculated as

$$\Delta E_{rev} = E_{rev,X} - E_{rev,K} = \frac{RT}{zF} \ln \frac{P_X}{P_K} \tag{3}$$

Values were compared between channel types and statistical significance assessed using ANOVA.

For calculating Rb$^+$/K$^+$ permeability (Fig. 3g), tail currents were elicited by a single pulse to +40 mV, followed by a pulse at −80 mV, from a holding potential of −80 mV in 98 mM [K$^+$] and then 98 mM [Rb$^+$]. The K$^+$ or Rb$^+$ conductance was calculated by dividing the peak amplitude of K$^+$ or Rb$^+$ carried tail currents by the driving force (difference between –80 mV, at which the tail currents are measured, and the equilibrium potential for K$^+$ or Rb$^+$ ions measured in the same oocyte).

Deactivating currents were fitted to a single-exponential standard function defined as follows:

$$\int(t) = \sum_{i=1}^{n} A_i e^{-t/\tau_i} + C. \tag{4}$$

**Chemical structures, in silico docking, and sequence analyses.** Chemical structures and electrostatic surface potentials (range, −0.1 to 0.1) were plotted using Jmol, an open-source Java viewer for chemical structures in 3D: http://jmol.org/. For docking, the $X.$ $laevis$ KCNQ1 cryo-EM structure[49] was first altered to incorporate KCNQ3/KCNQ5 residues known to be important for RTG and ML-213 binding, and their immediate neighbors, followed by energy minimization using the GROMOS 43B1 force field[50], in DeepView[51]. Thus, $X.$ $laevis$ KCNQ1 amino acid sequence LIT*TL*YIGF was converted to LIT*AW*YIGF, the underlined W being W265 in human KCNQ3/KCNQ5 and the italicized residues being the immediate neighbors in KCNQ3/KCNQ5. In addition, $X.$ $laevis$ KCNQ1 sequence WWG*VVTV*TTIGYGD was converted to WWG*LITLA*TIGYGD, the underlined L being Leu314 in human KCNQ3/KCNQ5 and the italicized residues being the immediate neighbors in KCNQ5 and/or KCNQ3. Surrounding non-mutated sequences are shown to illustrate the otherwise high sequence identity in these stretches. Unguided docking of mallotoxin and other compounds to predict binding sites was performed using SwissDock[52] with CHARMM forcefields[53].

**PTZ chemoconvulsant assay.** We compared anticonvulsant activities of test compounds in male C57BL/6 mice (Charles River) aged 2–3 months. Mice were housed and used according to the recommendations in the Guide for the Care and Use of Laboratory Animals of the National Institutes of Health (NIH Publication, 8th edition, 2011). The study protocol was approved by the Institutional Animal Care and Use Committee of University of California, Irvine, which confirmed that all relevant ethical regulations were adhered to. Chemicals were sourced from Sigma (St. Louis, MO, USA) . A pentylene tetrazole (PTZ) chemoconvulsant assay was used[54]. Mice were injected intraperitoneally with IVA and/or MTX, concentrations as indicated, or vehicle control (PBS with 1% DMSO), and then 30 min later injected intraperitoneally with 80 mg kg$^{-1}$ PTZ. Following the PTZ injection, mice were caged individually and an observer (GWA) blinded to the drug used recorded, over 20 min, clonic and tonic seizure incidence, and seizure-related mortality.

**Statistical analysis.** All values are expressed as mean ± SEM. Chi-squared analysis was used to compare seizure-related mortality in mice. One-way ANOVA was applied for all other tests; if multiple comparisons were performed, a post-hoc Tukey's HSD test was performed following ANOVA. All $p$-values were two-sided. Statistical significance was defined as $p < 0.05$.

## Data availability

The raw datasets generated during the current study are available from the corresponding author on reasonable request.

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

## Acknowledgements

This study was supported by the US National Institutes of Health (GM115189 to G.W. A.). We are grateful to Lily Chen and Angele De Silva (University of California, Irvine) for generating mutant channel constructs, and to Dr. Maria Papanikolaou (University of California, Irvine) for advice and technical assistance with seizure studies.

## Author contributions

R.W.M. conceived the study, performed the oocyte experiments and in silico docking, analyzed data, wrote a draft manuscript, and prepared figures. G.W.A. performed seizure

studies and in silico docking, analyzed seizure data, oversaw electrophysiological studies and suggested experiments, completed and edited the manuscript, and prepared figures.

## Additional information

**Competing interests:** The authors declare no competing interests.

