## [Peer Review File · Nature Communications]

Reviewers' comments:

Reviewer #1 (Remarks to the Author):

The authors analyzed in this study the effects of each ingredient included in *M. oppositifolius* leaf, an anticonvulsant herbal medicine, on KCNQ channels, and newly found MTX as well as IVA has a potentiation effect on KCNQ2 by shifting the G-V relationship to the hyperpolarized direction. They also identified the docking sites of MTX and IVA, based on mutagenesis study and in silico docking analysis. They observed synergistic effect of MTX and IVA. Furthermore, they found co-application of MTX, IVA with RTG, a known synthetic anticonvulsant, made the KCNQ2/3 channel constitutively active in all voltage ranges.

I very highly evaluate the general impact and scientific merits of this study. I judge the data quality is high and there is no major flaw. I have some comments, but most of them are for better understanding of the readers.

1. I wonder if the mechanistic interpretation using a state diagram (Fig. 6i, j) is necessary. Even without this part, this paper has sufficient scientific merits. In my opinion, this part is rather speculative with no strong experimental evidences. Thus, I think this part could be omitted and left for further intensive study in the future.

If the authors would like to include this part, I think it is necessary to add co-application data of KCNQ2 homo tetramer and KCNQ3 homo tetramer. It is because the assumed site of action, KCNQ2 subunit for MTX and IVA, and KCNQ3 subunit for RTG, is not necessarily clear. What kind of result is expected when the drugs are co-applied to KCNQ2 homo or KCNQ3 homo? Does the binding of MTX and IVA on KCNQ2 truly change the affinity for the binding of RTG on KCNQ3 as shown in Fig. 6j? How can the information of the binding of MTX and IVA on KCNQ2 be transmitted to the RTG binding site on KCNQ3?

2. The effect of 2-Mercaptophenol (2-MP) to shift the G-V relationship to the right is interesting. What will happen if it is co-applied with MTX, or with MTX & RTG. From the altogether application experiment in Fig. 5g, it is not clear.

3. I am curious to know the effect of muscarinic receptor mediated inhibition in the presence of these drugs, especially in the case of constitutively active one, KCNQ2/3 with MTX, IVA and RTG together.

4. A change in ion selectivity by MTX is shown in Fig. 3e, f, g. How about the ion selectivity of constitutively active current, when MTX, IVA and RTG are co-applied ?

5. All time constant data had better be plotted using a log scale. The significant difference at -120 mV is hard to see (Fig. 3 j.k). Also it would be better to present the tau values of activation and deactivation at, not one but e.g. three membrane potentials for better understanding.

6. The data points of WL/WL (orange) is hard to see (Fig. 3d).

Reviewer #2 (Remarks to the Author):

Thank you for the invitation to review 'Ancient and modern anticonvulsants act synergistically in a KCNQ potassium channel binding pocket', by Rian Manville and Geoffrey Abbott. This is an interesting paper that reports a significant amount of work identifying active components of a natural remedy (*Mallotus oppositifolius* leaves) that has been used to treat seizures. The authors report that a combination of two components (mallotoxin and isovaleric acid) act synergistically to activate KCNQ2/3 channels, and also to prevent seizures in a rodent model of PTZ-induced seizures. This effect is broken down into its individual components, analyzing the KCNQ isoform specificity, potency, and computational docking of each compound. Mallotoxin and isovaleric acid have distinct effects, and also notably distinct dependence on the presence of KCNQ2 W236 / KCNQ3 W265 – a residue that is essential for retigabine effects in these channels (it is required for isovaleric acid sensitivity, but not for mallotoxin). The most spectacular result in the paper is demonstrating that the combination of mallotoxin and isovaleric acid strongly sensitizes KCNQ2/3 channels to retigabine. This results in what appears to be a 'locked open' channel conformation, over the range of voltages tested (-120 mV - +40 mV). The paper is carefully written and straightforward to follow. Many aspects of the findings are interesting – especially the findings related to powerful actions on heteromeric (native) KCNQ2/3 channels. The main shortcoming of the paper is in the descriptions of possible mechanisms of action of the tested compounds and retigabine (see below). I think that re-examining how these mechanisms are conceived would help the paper, and increase its impact on our understanding of Kv7 channel pharmacology.

Major comments:

Line 135-136: "MTX speeded KCNQ2/3 activation and slowed deactivation, consistent with open state stabilization (Fig. 2d)." It was unclear to me why only open state stabilization was mentioned here exclusively, as opposed to an effect on multiple channel states. It seems clear from the recordings and the authors' statements that MTX speeds up channel activation, so a fair interpretation would be that the drug also destabilizes channel closed states.

Lines 188-208 – The authors use several comparisons to findings in reference 37, in order to distinguish the underlying mechanism of action of MTX and retigabine, particularly in terms of access/effect on various open states. Have the authors tried to recapitulate the findings described in reference 37 for retigabine, as a direct comparison in their experiments? The statement/implication that retigabine binds exclusively to a second open state does not seem consistent with some common observations of retigabine actions (most simply, as mentioned above, retigabine significantly accelerates channel activation – lacking some further explanation, the most reasonable mechanism is the drug also binds and destabilizes one or more pre-open states). Also, the concentrations of drugs used in reference 37 are very low and there is barely a shift reported in the voltage-dependence of activation. Overall, drawing a mechanistic distinction between MTX and RTG seemed tenuous based solely on these comparisons. A similar mechanistic argument is made

elsewhere in the paper (lines 277-278, lines 329-330), but alternative explanations should be considered.

In the cartoon model/mechanism (Fig. 6i,j) - the comment that the triple drug cocktail 'leverages subunit heterogeneity... a feature lacking in modern anticonvulsants' is correct in my opinion and is one of the more interesting aspects of this paper. However, given the comments above, I have reservations about the suggestion that MTX and IVA induce a state that allows retigabine to bind at negative voltages. I think a simple model in which the combined effects of the different compounds (likely on different subunits) generate a stable open conformation, is adequate and is better supported by the data. There must be something unique about the combination of the compounds, because binding of retigabine alone (also presumably to all four subunits in a KCNQ2/3) does not lock channels open after they are initially depolarized. A minor comment is that I did not find panel 6i to be very helpful at explaining the proposed model.

Lines 326-328: "When combined with the modern anticonvulsant RTG, the herbal extracts IVA and MTX have the unprecedented capacity to lock open KCNQ2/3 at all voltages tested, converting it into a voltage-independent "leak" channel." I agree these effects were very dramatic and unique. In terms of the phrasing of the statement - how stringently has this 'locked open' behavior been tested? More specifically, what is the interpulse interval and holding voltage? Is there a slow tail, with more prominent closure, if the interpulse interval is lengthened or more hyperpolarized? Some KCNQ openers cause dramatic slowing of channel closure of KCNQ2 channels (ztz240 or ICA-069673 in KCNQ2), although I agree that the effect on the KCNQ2/3 heteromeric channel is novel and unique.

Given the content of an interesting recent publication from these authors, I have to ask whether the reported effects of GABA or any of its metabolites are affected by the combination of IVA/MTX. This would be a really interesting addition to this paper, as it might relate to a physiological mechanism of action of the compounds.

Minor comments:

Fig 3d - perhaps it was a computer/display problem, but orange symbols were not visible?

Fig. 6 - Do you have any thoughts on why the 'triple-stimulated' (mtx+iva+retigabine) currents are blocked? Is this a vehicle effect (inhibition)?

There is a minor typo on line 106

Reviewer #3 (Remarks to the Author):

This is an interesting paper that investigates the potential mechanisms of action of two compounds found in an extract of a herbal therapy traditionally used to treat epilepsy.

From the clinical point of view, this information is important as it provides a potential rationale for how the extract may exert clinical effects, and it also points out the synergistic effect of the combination of the compounds, which is important from the standpoint of drug development.

While not critically important to the aims of the paper or its conclusions, additional information about MTX and IVA would be desirable to further raise clinical interest:

1. Is there any evidence that MTX and IVA cross the blood-brain barrier besides the rodent epilepsy experiments, and if so, have areas in brain that show binding been identified?

2. Are there any data that estimate how much MTX and IVA would be found in the dosage of extract that a person might ingest? How much would be absorbed through the GI tract? Whether the amounts that cross the blood-brain barrier would reach sufficient concentration at KCNQ2/3 channels to have the effect documented in the paper? Or conversely, can you go backwards by estimating what the oral dosages of MTX and IVA would need to be in order that these compounds achieve sufficient concentrations at their target?

Manville and Abbott – Response to Reviewers

My co-author and I are extremely grateful to the reviewers for their enthusiasm and their thoughtful feedback. We have addressed the comments with three new figures containing results from several new experiments, and rewriting of the manuscript. The changes and additions are described below in a point-by-point response to the reviewers' comments, and are highlighted in the manuscript. Please note that data from the new experiments are also summarized in new Supplementary Tables 10, 11, 19-21, 30, 32, 33, 36-39 in the Extended Data.

Reviewers' comments:

Reviewer #1 (Remarks to the Author):

The authors analyzed in this study the effects of each ingredient included in *M. oppositifolius* leaf, an anticonvulsant herbal medicine, on KCNQ channels, and newly found MTX as well as IVA has a potentiation effect on KCNQ2 by shifting the G-V relationship to the hyperpolarized direction. They also identified the docking sites of MTX and IVA, based on mutagenesis study and in silico docking analysis. They observed synergistic effect of MTX and IVA. Furthermore, they found co-application of MTX, IVA with RTG, a known synthetic anticonvulsant, made the KCNQ2/3 channel constitutively active in all voltage ranges.

I very highly evaluate the general impact and scientific merits of this study. I judge the data quality is high and there is no major flaw. I have some comments, but most of them are for better understanding of the readers.

Thank you.

1. I wonder if the mechanistic interpretation using a state diagram (Fig. 6i, j) is necessary. Even without this part, this paper has sufficient scientific merits. In my opinion, this part is rather speculative with no strong experimental evidences. Thus, I think this part could be omitted and left for further intensive study in the future.

If the authors would like to include this part, I think it is necessary to add co-application data of KCNQ2 homo tetramer and KCNQ3 homo tetramer. It is because the assumed site of action, KCNQ2 subunit for MTX and IVA, and KCNQ3 subunit for RTG, is not necessarily clear. What kind of result is expected when the drugs are co-applied to KCNQ2 homo or KCNQ3 homo? Does the binding of MTX and IVA on KCNQ2 truly change the affinity for the binding of RTG on KCNQ3 as shown in Fig. 6j? How can the information of the binding of MTX and IVA on KCNQ2 be transmitted to the RTG binding site on KCNQ3?

We have removed the state diagram because it was deemed unnecessary for the major impact of the manuscript. We have also simplified the mechanistic interpretation to avoid unnecessary speculation. In addition, as the reviewer requested, we have also now quantified the effects of the different combinations and concentrations of the drugs on homomeric KCNQ channels (isoforms 2 – 5). These data are shown in a new Figure 9 (panels a-d), together with a direct comparison of drug effects during extended periods at -120 mV (Figure 9e,f), the simplified model (Figure 9g) and the original docking results (Figure 9h,i) that we now feel reflect our data without over-interpreting, in response to the

reviewer's requests. **The homomeric data show that while KCNQ3 is overall the most sensitive isoform to the combination, it still cannot be locked open in the manner that KCNQ2/3 heteromers are, reinforcing our idea that the combination leverages the heteromeric channel composition.**

2. The effect of 2-Mercaptophenol (2-MP) to shift the G-V relationship to the right is interesting. What will happen if it is co-applied with MTX, or with MTX & RTG. From the altogether application experiment in Fig. 5g, it is not clear.

To address the reviewer's comments and interest we have conducted more experiments and also docking, the results of which we show in a new Figure 4. First, we show that GABOB, which we previously found to be a high-affinity partial agonist for W265 on KCNQ3, only partially competes out MTX, as one would predict from docking and mutagenesis studies that suggest GABOB binds more closely than MTX to W265, albeit in a similar binding pocket (Figure 4a-d). Next we show that 2-Mercaptophenol docks close to W265 in silico, and that mutation of W236 and W265 renders KCNQ2/3 channels insensitive to 2-Mercaptophenol (Figure 4e-g). Finally, we show that 100 μ M 2-Mercaptophenol is ineffective at competing out 30 μ M MTX. This would be predicted because 2-Mercaptophenol is much more reliant than MTX on W265 and is predicted to bind more closely to W265 than does MTX; further, while GABOB is a high affinity partial agonist ($EC_{50} = 120$ nM), 2-Mercaptophenol only begins to inhibit at ~ 100 μ M, and is an activator at lower concentrations.

3. I am curious to know the effect of muscarinic receptor mediated inhibition in the presence of these drugs, especially in the case of constitutively active one, KCNQ2/3 with MTX, IVA and RTG together.

This is a very interesting question to address. We plan to examine this in detail in a separate study, together with the study on GABA and metabolites suggested by Reviewer 2.

4. A change in ion selectivity by MTX is shown in Fig. 3e, f, g. How about the ion selectivity of constitutively active current, when MTX, IVA and RTG are co-applied ?

We have now conducted these experiments for both heteromeric KCNQ2/3 and homomeric KCNQ2 and KCNQ3 channels, and report the results in a new Figure 8. We show that MTX/IVA/RTG induces even higher Na^+ and Cs^+ permeability than we saw for MTX alone, and it does it for all three channels. Together with data in Figures 7 and 9, this suggests that all three channels undergo somewhat similar conformational changes in response to the combination, but that KCNQ2/3 is able to undergo the transition at more hyperpolarized membrane potentials.

5. All time constant data had better be plotted using a log scale. The significant difference at -120 mV is hard to see (Fig. 3 j.k). Also it would be better to present the tau values of activation and deactivation at, not one but e.g. three membrane potentials for better understanding.

We have added data for multiple voltages and also now plot the time constant data on log scales.

6. The data points of WL/WL (orange) is hard to see (Fig. 3d).

We have made the symbols open and one of the lines dashed so that the orange data points are not obscured by the purple, which were largely overlapping.

Reviewer #2 (Remarks to the Author):

Thank you for the invitation to review ‘Ancient and modern anticonvulsants act synergistically in a KCNQ potassium channel binding pocket’, by Rian Manville and Geoffrey Abbott. This is an interesting paper that reports a significant amount of work identifying active components of a natural remedy (*Mallotus oppositifolius* leaves) that has been used to treat seizures. The authors report that a combination of two components (mallotoxin and isovaleric acid) act synergistically to activate KCNQ2/3 channels, and also to prevent seizures in a rodent model of PTZ-induced seizures. This effect is broken down into its individual components, analyzing the KCNQ isoform specificity, potency, and computational docking of each compound. Mallotoxin and isovaleric acid have distinct effects, and also notably distinct dependence on the presence of KCNQ2 W236 / KCNQ3 W265 – a residue that is essential for retigabine effects in these channels (it is required for isovaleric acid sensitivity, but not for mallotoxin). The most spectacular result in the paper is demonstrating that the combination of mallotoxin and isovaleric acid strongly sensitizes KCNQ2/3 channels to retigabine. This results in what appears to be a ‘locked open’ channel conformation, over the range of voltages tested (-120 mV - +40 mV).

The paper is carefully written and straightforward to follow. Many aspects of the findings are interesting – especially the findings related to powerful actions on heteromeric (native) KCNQ2/3 channels. The main shortcoming of the paper is in the descriptions of possible mechanisms of action of the tested compounds and retigabine (see below). I think that re-examining how these mechanisms are conceived would help the paper, and increase its impact on our understanding of Kv7 channel pharmacology.

Thank you – we have performed new experiments, added 3 new figures and edited the manuscript to address these constructive comments.

Major comments:

Line 135-136: “MTX speeded KCNQ2/3 activation and slowed deactivation, consistent with open state stabilization (Fig. 2d).” It was unclear to me why only open state stabilization was mentioned here exclusively, as opposed to an effect on multiple channel states. It seems clear from the recordings and the authors’ statements that MTX speeds up channel activation, so a fair interpretation would be that the drug also destabilizes channel closed states.

As requested, we have amended this statement to now indicate that the drug also destabilizes channel closed states.

Lines 188-208 – The authors use several comparisons to findings in reference 37, in order to distinguish the underlying mechanism of action of MTX and retigabine, particularly in terms of access/effect on various open states. Have the authors tried to recapitulate the findings described in reference 37 for retigabine, as a direct comparison in their experiments? The

statement/implication that retigabine binds exclusively to a second open state does not seem consistent with some common observations of retigabine actions (most simply, as mentioned above, retigabine significantly accelerates channel activation – lacking some further explanation, the most reasonable mechanism is the drug also binds and destabilizes one or more pre-open states). Also, the concentrations of drugs used in reference 37 are very low and there is barely a shift reported in the voltage-dependence of activation. Overall, drawing a mechanistic distinction between MTX and RTG seemed tenuous based solely on these comparisons. A similar mechanistic argument is made elsewhere in the paper (lines 277-278, lines 329-330), but alternative explanations should be considered. In the cartoon model/mechanism (Fig. 6i,j) - the comment that the triple drug cocktail ‘leverages subunit heterogeneity... a feature lacking in modern anticonvulsants’ is correct in my opinion and is one of the more interesting aspects of this paper. However, given the comments above, I have reservations about the suggestion that MTX and IVA induce a state that allows retigabine to bind at negative voltages. I think a simple model in which the combined effects of the different compounds (likely on different subunits) generate a stable open conformation, is adequate and is better supported by the data. There must be something unique about the combination of the compounds, because binding of retigabine alone (also presumably to all four subunits in a KCNQ2/3) does not lock channels open after they are initially depolarized. A minor comment is that I did not find panel 6i to be very helpful at explaining the proposed model.

We have taken these comments on board, and removed references to reference 37 and also the models pertaining to this mechanism. Instead, as the reviewer now suggests, we focus on a simple model in which the combination of compounds generates a stable open state. We have removed 6i as suggested, and also simplified 6j (now Figure 9g) to reflect the reviewer’s ideas.

Lines 326-328: “When combined with the modern anticonvulsant RTG, the herbal extracts IVA and MTX have the unprecedented capacity to lock open KCNQ2/3 at all voltages tested, converting it into a voltage-independent “leak” channel.” I agree these effects were very dramatic and unique. In terms of the phrasing of the statement - how stringently has this ‘locked open’ behavior been tested? More specifically, what is the interpulse interval and holding voltage? Is there a slow tail, with more prominent closure, if the interpulse interval is lengthened or more hyperpolarized? Some KCNQ openers cause dramatic slowing of channel closure of KCNQ2 channels (ztz240 or ICA-069673 in KCNQ2), although I agree that the effect on the KCNQ2/3 heteromeric channel is novel and unique.

We have added new data from a voltage protocol with a 20 s -120 mV pulse to show that the RTG +MTX/IVA combination locks KCNQ2/3 open for extended periods at -120 mV (Figure 9e,f), in sharp contrast to homomeric KCNQ3 under similar conditions. We have also added a full characterization of the effects of the various drug concentrations and combinations on homomeric KCNQ2-5 channels for a full comparison (Figure 9a-d). Together, the data show that KCNQ2/3 is unique in being highly resistant to deactivation in the presence of the triple drug combination.

Given the content of an interesting recent publication from these authors, I have to ask whether the reported effects of GABA or any of its metabolites are affected by the combination of IVA/MTX. This would be a really interesting addition to this paper, as it might relate to a physiological mechanism of action of the compounds.

This is indeed a very interesting question to address. We plan to begin a new study examining this in detail, which we consider outside the scope of the present study as it will take a great many different concentrations of the drugs and metabolites to assess comprehensively. However, we have added new data showing that GABOB, a partial agonist, can interfere with IVA and to a lesser extent MTX effects (new Figure 4a-d and Figure 5j-l).

Minor comments:

Fig 3d – perhaps it was a computer/display problem, but orange symbols were not visible?

We have made the symbols open and one of the lines dashed so that the orange data points are not obscured by the purple, which were largely overlapping at several values.

Fig. 6 – Do you have any thoughts on why the ‘triple-stimulated’ (mtx+iva+retigabine) currents are blocked? Is this a vehicle effect (inhibition)?

We do not think this is a vehicle effect as the vehicle is similar for MTX+RTG with or without IVA, for example. One possible explanation is that it arises from the altered ion selectivity, which we show in Figure 8.

There is a minor typo on line 106

We have corrected this, thanks.

Reviewer #3 (Remarks to the Author):

This is an interesting paper that investigates the potential mechanisms of action of two compounds found in an extract of a herbal therapy traditionally used to treat epilepsy.

From the clinical point of view, this information is important as it provides a potential rationale for how the extract may exert clinical effects, and it also points out the synergistic effect of the combination of the compounds, which is important from the standpoint of drug development.

While not critically important to the aims of the paper or its conclusions, additional information about MTX and IVA would be desirable to further raise clinical interest:

1. Is there any evidence that MTX and IVA cross the blood-brain barrier besides the rodent epilepsy experiments, and if so, have areas in brain that show binding been identified?

Volatile fatty acids similar to IVA are known to cross the blood brain barrier. We have added this note to paragraph 2 of the Discussion. The binding studies have not been performed, to our knowledge.

2. Are there any data that estimate how much MTX and IVA would be found in the dosage of extract that a person might ingest? How much would be absorbed through the GI tract? Whether the amounts that cross the blood-brain barrier would reach sufficient concentration at KCNQ2/3

channels to have the effect documented in the paper? Or conversely, can you go backwards by estimating what the oral dosages of MTX and IVA would need to be in order that these compounds achieve sufficient concentrations at their target?

We have added a summary of the available data on this, to paragraphs 1 & 2 of the Discussion.

REVIEWERS' COMMENTS:

Reviewer #1 (Remarks to the Author):

The revisions by the authors are truly intensive, including the addition of three new Figures 9, 4 and 8 and careful rewriting. I am fully satisfied with the revisions. I have no more specific concern and would like to recommend acceptance in the present form. I believe this paper has a very high impact to the wide-ranged readers of Nature Communications.

Reviewer #2 (Remarks to the Author):

I have no further comments on the manuscript. Thanks for addressing all comments.

Reviewer #3 (Remarks to the Author):

The authors have satisfactorily addressed my concerns.

Response to reviewers

REVIEWERS' COMMENTS:

Reviewer #1 (Remarks to the Author):

The revisions by the authors are truly intensive, including the addition of three new Figures 9, 4 and 8 and careful rewriting. I am fully satisfied with the revisions. I have no more specific concern and would like to recommend acceptance in the present form. I believe this paper has a very high impact to the wide-ranged readers of Nature Communications.

Reviewer #2 (Remarks to the Author):

I have no further comments on the manuscript. Thanks for addressing all comments.

Reviewer #3 (Remarks to the Author):

The authors have satisfactorily addressed my concerns.

My co-author and I greatly appreciate all three reviewers' careful work and enthusiasm for this manuscript.